# Lack of evidence for increased transcriptional noise in aged tissues

Olga Ibañez-Solé[1,2†], Alex M Ascensión[1,2†], Marcos J Araúzo-Bravo[1,3,4,5]*, Ander Izeta[2,6]*

[1]Biodonostia Health Research Institute, Computational Biology and Systems Biomedicine Group, Donostia-San Sebastián, Spain; [2]Biodonostia Health Research Institute, Tissue Engineering group, Donostia-San Sebastián, Spain; [3]Biodonostia Health Research Institute, Computational Biomedicine Data Analysis Platform, Donostia-San Sebastián, Spain; [4]CIBER of Frailty and Healthy Aging (CIBERfes), Madrid, Spain; [5]IKERBASQUE, Basque Foundation for Science, Bilbao, Spain; [6]Tecnun-University of Navarra, Donostia-San Sebastián, Spain

**Abstract** Aging is often associated with a loss of cell type identity that results in an increase in transcriptional noise in aged tissues. If this phenomenon reflects a fundamental property of aging remains an open question. Transcriptional changes at the cellular level are best detected by single-cell RNA sequencing (scRNAseq). However, the diverse computational methods used for the quantification of age-related loss of cellular identity have prevented reaching meaningful conclusions by direct comparison of existing scRNAseq datasets. To address these issues we created *Decibel*, a Python toolkit that implements side-to-side four commonly used methods for the quantification of age-related transcriptional noise in scRNAseq data. Additionally, we developed *Scallop*, a novel computational method for the quantification of membership of single cells to their assigned cell type cluster. Cells with a greater *Scallop* membership score are transcriptionally more stable. Application of these computational tools to seven aging datasets showed large variability between tissues and datasets, suggesting that increased transcriptional noise is not a universal hallmark of aging. To understand the source of apparent loss of cell type identity associated with aging, we analyzed cell type-specific changes in transcriptional noise and the changes in cell type composition of the mammalian lung. No robust pattern of cell type-specific transcriptional noise alteration was found across aging lung datasets. In contrast, age-associated changes in cell type composition of the lung were consistently found, particularly of immune cells. These results suggest that claims of increased transcriptional noise of aged tissues should be reformulated.

**\*For correspondence:**
mararabra@yahoo.co.uk (MJA-B);
ander.izeta@biodonostia.org (AI)

[†]These authors contributed equally to this work

**Competing interest:** The authors declare that no competing interests exist.

## Editor's evaluation

The authors present an important perspective surrounding a fundamental question of associations between transcriptional noise and the aging process. They develop new methods to probe stochastic gene expression from single-cell sequencing data where their results suggest that associations between noise and age can be attributed to alternative metrics such as shifts in cellular identity. These methods and analyses provide an important framework to guide the fields of gene expression regulation and aging.

## Introduction

Concomitant to the large repertoire of known age-associated changes at the cellular level, an increase in transcriptional variability is generally assumed to characterize aged cells and tissues (*Nikopoulou*

**eLife digest** The human body contains hundreds of different cell types which vary greatly in shape and size despite all sharing the same genetic material. This is because each cell switches on, or 'expresses', a unique set of genes that gives them a specific identity, such as becoming a nerve or a muscle cell.

Recent studies have shown that cells in some tissues tend to lose their identity with age, and activate some of the genes that define them less strongly. This results in seemingly identical cells expressing the same genes in a more variable way, a phenomenon commonly referred to as noise.

A technique called single-cell RNA sequencing is typically used to measure the activity of genes in individual cells, and has been used to study the role of noise in a wide range of aging tissues. However, the results of these studies have been analyzed using different computational methods, making it difficult to make comparisons between tissues and organisms. This has led to an ongoing debate about whether increased noise is a signature feature of aging, and if it is experienced throughout the body or restricted to certain cell types.

To overcome this, Ibáñez-Solé, Ascensión et al. developed two new computational tools for analyzing noise and changes in cell identity: these were then applied to seven unique sequencing datasets which had been collected from various tissues in humans and mice at different ages.

While there were some differences in the level of noise between young and old cells, these changes were not consistent across tissues and organisms. In contrast, other features associated with aging were consistently found in each of the sequencing datasets.

The role of noise in aging has been gaining increasingly more attention in the scientific literature. However, the findings of Ibáñez-Solé, Ascensión et al. suggest that this phenomenon is not a hallmark of the aging process, and that the field should focus on other factors that reduce the health of tissues and cells as organisms get older. The computational approach they developed could also be used to evaluate the role of noise in other contexts, such as certain diseases.

*et al., 2019*; *Uyar et al., 2020*; *Mendenhall et al., 2021*; *Vijg, 2021*). This phenomenon was first described by Vijg and colleagues as an *age-related increase in transcriptional noise* (*Bahar et al., 2006*), which is still the most commonly used term (*Warren et al., 2007*; *Enge et al., 2017*; *Angelidis et al., 2019*). *Transcriptional noise* is here defined as the measured level of variation in gene expression among cells supposed to be identical (*Raser and O'Shea, 2005*). Later, similar findings have been reported as an increase in *identity noise* (*Salzer et al., 2018*), *cell-cell heterogeneity* (*Kimmel et al., 2019*), *cell-to-cell variability* (*Martinez-Jimenez et al., 2019*; *Ximerakis et al., 2019*; *Hernando-Herraez et al., 2019*), or *loss of cellular identity* in aged tissues (*Solé-Boldo et al., 2020*). However, a loss of cellular identity does not necessarily concur with an increase in transcriptional noise, and both phenomena are thus best studied separately. Additionally, while all these claims have in common the notion of cells expressing their core transcriptional program or *transcriptomic signature* in a loose way, there are important methodological differences between the published reports that deserve further scrutiny.

Early studies were based on the quantification of the variance associated with the expression of a few pre-selected transcripts by real-time PCR, on bulk cell and tissue samples (*Bahar et al., 2006*; *Warren et al., 2007*). With the advent of single-cell RNA sequencing (scRNAseq) technologies, whole-transcriptome variability on aged tissues was studied at the single-cell level. A pioneering study on human pancreas by Quake and colleagues found an age-related increase in transcriptional noise specific to pancreatic $\beta$ cells (*Enge et al., 2017*). The authors introduced a definition of transcriptional noise that was based on whole-transcriptome variability: the ratio between biological and technical variation, where the latter was inferred from External RNA Controls Consortium (ERCC) spike-in variability. As ERCC spike-in controls are not included in every scRNAseq experiment, they proposed two alternative methods that were based on the notion of 'distance to centroid (DTC)': the greater the gene-based distance between cells of the same cell type, the greater the transcriptional noise associated with them. One of them measured the Euclidean distance to the cell type mean per individual, using the whole transcriptome. The other one measured the Euclidean distance between each cell and the tissue mean, using a set of invariant genes. Soon after, loss of identity was reported in aged

murine dermal fibroblasts by measuring the coefficient of variation of the distances between each highly variable gene between the two main fibroblast clusters (*Salzer et al., 2018*). Similar findings were published in early activated CD4+ T cells, based on the observation that the fraction of cells that expressed the core activation program was lower in old animals (*Martinez-Jimenez et al., 2019*). A study on murine aging lung found an increase in cellular heterogeneity in most (but not all) cell types (*Angelidis et al., 2019*), based on the distance-to-mean method of *Enge et al., 2017*. Later, a study on murine lung, spleen, and kidney corroborated by Euclidean DTC methods an age-related increase in cell-to-cell variability, albeit present in some cell types only (*Kimmel et al., 2019*). In contrast, a study in murine aging brain found no increase in transcriptional heterogeneity associated with aging (*Ximerakis et al., 2019*). Overall, these results suggest that the purported age-associated increase in transcriptional noise might be restricted to particular cell types or tissues.

Of note, alternative explanations to the variability in the expression of individual genes being the basis for increased transcriptional noise do exist. Among others, the lack of balance between spliced and unspliced mRNAs (*Gupta et al., 2021*) and the existence of dysregulated gene regulatory networks (*Mishra et al., 2021*) have been proposed. In fact, Bashan and colleagues developed a novel computational tool to measure age-related loss of gene-to-gene transcriptional coordination (what they called *global coordination level* or GCL), and reported a GCL decrease in aging cells across diverse organisms and cell types, which was also associated with a high mutational load (*Levy et al., 2020*). In a nutshell, these authors suggested that the observations of age-associated increase in cell-to-cell variability were restricted to specific cell types and tissues but not generalized. Instead, they proposed that transcriptional dysregulation occurs at the level of gene-to-gene coordination. Despite the numerous attempts at measuring transcriptional noise in aged tissues, several challenges remain: (i) there are important differences in between studies with regard to the definition of *transcriptional noise* and the computational methods used to quantify it; (ii) studies focused mostly on single datasets of different tissues and cell types, while it is well known that both the inter-tissue and the inter-cellular variability might be significant; and (iii) little to no attention was given to the fact that cellular composition of aged organs shows relevant variability as compared to the young (*Nalapareddy et al., 2022*).

In the present work, we aimed to systematically measure age-associated transcriptional noise across different tissues and species, testing diverse computational methods in parallel. The main goals of the study were to substantiate claims of age-associated transcriptional noise increase and determine whether it presents a cell type-specific pattern. For this, we took advantage of the large number of aging mouse and human scRNAseq datasets that are publicly available and developed two computational tools (*Decibel* and *Scallop*) to analyze them by focusing on two aspects: age-related transcriptional noise and changes in cell type composition.

## Results

### *Decibel*: a Python toolkit for transcriptional noise quantification

We developed a Python toolkit for the quantification of transcriptional noise in scRNAseq, where we implemented the four main families of methods that have been used in the literature to measure increase in transcriptional noise associated with aging (*Figure 1A*). The first method, which we refer to as *biological variation over technical variation*, takes the Pearson's correlation distance between each cell and the mean expression vector of its corresponding cell type for that individual, using the whole transcriptome in the calculation. It then divides this correlation by the ERCC-based distance between each cell and its cell type mean. This method can only be used when ERCC spike-ins have been included in the experimental design. The second method computes the gene-based Euclidean distance between each cell's expression vector and its cell type mean expression vector per donor/individual. The third one computes the Euclidean distance between each cell and the average gene expression across cell types, using a set of invariant genes. Invariant genes are selected by splitting the whole transcriptome into 10 equally sized bins according to their mean expression. Then, the two bins with the most extreme expression values are discarded, and the 10% with the highest coefficient of variation within each of the remaining bins are selected. The fourth one is the GCL. Its original formulation takes a dataset containing a single cell type, and it randomly splits its transcriptome into two halves, then computes the dependency between them as the batch-corrected distance correlation (*Levy et al., 2020*). The GCL is obtained by averaging this dependency over *k* iterations. We

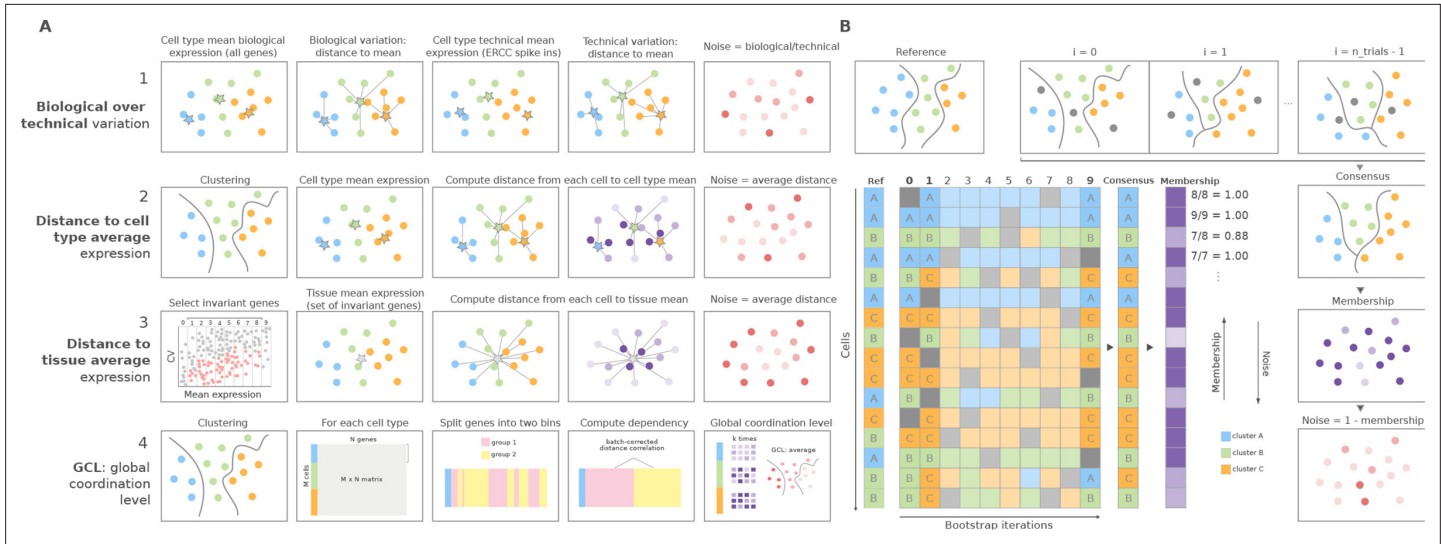

**Figure 1.** Overview of computational methods for the quantification of transcriptional noise and example workflow in *Scallop*. (**A**) The methods implemented in *Decibel* Python toolkit are summarized through diagrams depicting how they measure transcriptional noise. (1) Biological variation (whole transcriptome-based Pearson's correlation distance between each cell and the mean expression vector), divided by the technical variation (External RNA Controls Consortium [ERCC] spike-in based distance; *Enge et al., 2017*). (2) Mean whole transcriptome-based Euclidean distance to cell type average (*Enge et al., 2017*). (3) Mean invariant gene-based Euclidean distance to tissue average (*Enge et al., 2017*). (4) GCL (*Levy et al., 2020*) per cell type. Stars represent the 'center' of each cluster (average gene expression for each cell type). (**B**) *Scallop*: example workflow on a 16 cell dataset. A reference clustering solution (*Ref*) is obtained by running a community detection algorithm (default: Leiden) on the whole dataset. Three clusters are obtained: A (blue), B (green), and C (orange). Then, a subset of cells is randomly selected and subjected to unsupervised clustering *n_trials* = 10 times (cells not selected in each bootstrap iteration are shown in gray). The cluster labels across bootstrap iterations are harmonized by mapping the cluster labels with the greatest overlap, using the Hungarian method (*Munkres, 1957*). A consensus clustering solution is derived by selecting the most frequently assigned cluster label per cell, and the membership score is computed as the frequency with which the consensus label was assigned to each cell. *Scallop* measures noise as a *1 − membership* value assigned to each cell.

The online version of this article includes the following figure supplement(s) for figure 1:

**Figure supplement 1.** Performance of *Scallop* and two distance-to-centroid methods on four artificial datasets with increasing transcriptional noise.

**Figure supplement 2.** Ability of *Scallop* and a distance-to-centroid method to detect noisy cells within cell type clusters.

**Figure supplement 3.** Effect of cellular composition on the performance of *Scallop*.

**Figure supplement 4.** Effect of dataset size on the performance of *Scallop*.

**Figure supplement 5.** Effect of the number of genes on the performance of *Scallop*.

**Figure supplement 6.** Effect of marker expression on the performance of *Scallop*.

**Figure supplement 7.** Performance of *Scallop* in comparison to pre-existing methods for the quantification of transcriptional noise.

**Figure supplement 8.** *Scallop* robustness in relation to input parameters.

**Figure supplement 9.** Stable cells as identified with *Scallop* are more representative of the cell type than unstable cells.

implemented an extension of this method so that it could be used in datasets containing several cell types, by computing the GCL averaged over 50 iterations for each cell type of the same individual. Therefore, our implementation outputs a GCL score per cell type and individual rather than a transcriptional variability measure per cell. The Python implementation of these four methods is available at https://gitlab.com/olgaibanez/decibel, (copy archived at swh:1:rev:8749a4e1ae05edcebb642f-d7358a78b8468c511f; *Ibañez-Solé, 2022a*).

## *Scallop* membership score accurately identifies transcriptionally noisy cells

In addition to implementing existing methods, we developed *Scallop*, a novel tool for the quantification of the degree of loss of cell type identity in scRNAseq data (*Figure 1B*). *Scallop* measures the membership of each cell to a particular cluster by iteratively running a clustering algorithm on randomly selected subsets of cells and computing the fraction of iterations a cell was assigned to

a particular cluster. Thus, cluster membership takes values between 0 and 1. *Scallop* relies on the assumption that the more consistently a cell is assigned to a particular cluster across bootstrap iterations, the greater its transcriptional stability. Conversely, a cell being assigned to different clusters across iterations is indicative of a greater transcriptional variation. Therefore, we quantify loss of cell type identity as $1 - membership$. A detailed description of the three steps of the method (bootstrapping, cluster relabeling, and computation of the membership score) is provided in the *Scallop* subsection in the Methods.

In order to characterize and validate our method for transcriptional noise quantification, we conducted three types of analyses. First, we used artificially generated datasets containing various degrees of transcriptional noise to compare the performance of *Scallop* and DTC methods side by side, regarding their ability to measure transcriptional noise and detect noisy cells within cell types. Next, we ran simulations using artificial datasets in order to study the effect of a number of dataset features on the performance of *Scallop*: cellular composition, dataset size, number of genes, and marker expression. Finally, we graphically evaluated the output of *Scallop* on a dataset of human T cells, we analyzed its robustness to its input parameters, and we studied the relationship between membership and robust marker expression, using the PBMC 3 K dataset from 10× Genomics.

We compared the output of *Scallop* and two DTC methods (the whole transcriptome-based Euclidean distance to average cell type expression and the invariant gene-based Euclidean distance to average tissue expression) on four artificially generated datasets containing various levels of transcriptional noise. The analysis showed that *Scallop*, unlike DTC methods, was able to discern between the core transcriptionally stable cells within each cell type cluster from the more noisy cells that lie in between clusters (see *Figure 1—figure supplement 1*). We then compared one of the DTC methods to *Scallop* regarding their ability to detect noisy cells within each of the cell types, by plotting the top 10% noisiest and top 10% most stable cells (see *Figure 1—figure supplement 2A*). Analyzing the distribution of noise values for each cell type separately revealed that *Scallop* can distinguish between clusters that mainly consist of transcriptionally stable cells from noisier clusters that do not have such a distinct transcriptional signature (*Figure 1—figure supplement 2B*).

Next, we ran a series of simulations on artificially generated datasets to evaluate the performance of *Scallop* in the presence of different levels of class imbalance, dataset size, number of genes, and different degrees of expression of cell type markers. Our analysis showed that *Scallop* was remarkably robust to changes in cellular composition (see *Figure 1—figure supplement 3*). Both the average percentage of noise and the distribution remained unchanged for a wide range of class imbalance degrees (IDs). Similarly, altering the dataset size (number of cells) and the number of genes of an artificial dataset did not cause any major changes on the transcriptional noise values attributed to each cell type (see *Figure 1—figure supplements 4 and 5*). Additionally, we conducted an analysis where we identified the 10 most differentially expressed gene (DEG) markers for a cell type and measured the transcriptional noise associated with that cell type as we removed the expression of those genes from the dataset (*Figure 1—figure supplement 5*). Transcriptional noise steadily increased as we removed the effect of the top marker genes that defined the cell type under study (see *Figure 1—figure supplement 5B*). This experiment provides further evidence on how strong marker expression is related to robust cell type identity and how the lack of it results in transcriptional noise.

We extended the validation of our method to biological datasets: we compared the output of *Scallop* to the transcriptional noise measured using the methods implemented in *Decibel* on 8278 human T cells drawn from the Peripheral Blood Mononuclear Cell (PBMC) 20 K dataset from 10× Genomics. First, clustering revealed three main T cell subtypes, which we annotated according to their expression of *CD4* and *CD8* markers (*Figure 1—figure supplement 7A*). Then, we measured transcriptional variability and gene coordination level using *Decibel* and inspected the distribution of variability scores over the uniform manifold approximation and projection (UMAP) plots (*McInnes et al., 2018*; *Figure 1—figure supplement 1Figure 1—figure supplement 7B*). Unlike DTC methods, *Scallop* detected transcriptionally noisy cells that lie in between transcriptionally stable T cell subtypes on the UMAP plot. GCL yielded different coordination levels for each T cell subtype, but the method does not allow for comparisons between individual cells, as it outputs a single score per cell type. In addition, we plotted the 10% most transcriptionally stable and unstable cells according to the *Euclidean distance to the cell type mean* and *Scallop* methods (*Figure 1—figure supplement 7C*). These analyses suggested that *Scallop*'s membership score outperforms DTC methods at

discriminating between noisy cells lying in between clusters and more transcriptionally robust cells constituting the core of T cell subtypes.

Next, we analyzed *Scallop*'s robustness in response to input parameters, namely, the number of bootstrap iterations and the fraction of cells used in each iteration. We ran *Scallop* on five independent scRNAseq datasets with different size and cluster composition (see Appendix 1) and studied the convergence of *Scallop* membership scores for a wide range of values (*Figure 1—figure supplement 8*). The median correlation distance between membership scores decreased as we increased the number of bootstrap iterations (`n_trials`) and the fraction of cells used in each iteration (`frac_cells`). We concluded that *Scallop*'s output is robust to changes in its input parameter values, the results suggesting that `frac_cells` >0.8 and `n_trials` >30 are appropriate parameter values for most datasets (*Figure 1—figure supplement 8*).

Finally, we studied the relationship between *Scallop* membership score and robust gene marker expression, by comparing the statistical significance of the output of differential expression analysis between cell type clusters, conducted on stable and unstable cells. For this, we analyzed six cell types and subtypes (*CD4* and *CD8* T cells, *CD14* and *FCGR3A* monocytes, dendritic cells, and natural killer [NK] cells) from the PBMC 3 K dataset from 10× Genomics. Cells with a higher *Scallop* membership to their cluster differentially expressed cell type-specific markers with greater statistical significance than low-membership cells (*Figure 1—figure supplement 9*). Overall, these results showed that *Scallop* membership is related to a more robust expression of gene markers defining cell types than other existing methods.

## Increased transcriptional noise is not a universal hallmark of aging

To determine if aging is associated with a generalized increase in transcriptional noise at the tissue level, we used *Scallop* to compare the average degree of membership of young and old cells to their cell type cluster in scRNAseq datasets of various tissues (*Figure 2*). For the initial analysis, we selected seven datasets where transcriptional noise had already been measured using different methods and with differing outcomes. We provide a summary of the main characteristics of each dataset, as well as the findings regarding transcriptional noise obtained in each of the original studies, whether changes in transcriptional noise were restricted to particular cell types and the computational method used to measure noise (see Appendix 2). The age and cell type composition of the final datasets used in our study are shown in *Figure 2—figure supplement 1*, and the samples included in the datasets as well as the inclusion criteria are provided in Appendix 3 . Additionally, the methods implemented in *Decibel* to compute loss of identity were run in parallel as a control (*Figure 2—figure supplement 2*). When measuring the *Scallop* membership score of individual young and old cells to their cell type clusters, the results were inconsistent. Differences between age groups were found in some datasets, but the directionality of the change was not conserved across datasets, neither in the average $1 - membership$ score nor in the percentage of noisy cells in the young and the old fraction of each dataset. For most datasets (*Angelidis et al., 2019*; *Ximerakis et al., 2019*; *Kimmel et al., 2019*; *Martinez-Jimenez et al., 2019*), no significant change in mean transcriptional noise was found. Two datasets (*Enge et al., 2017*; *Salzer et al., 2018*) showed an increase in mean membership associated with aging, although we observed the interquartile range of noise values to be very similar between young and old individuals. In one of the datasets (*Solé-Boldo et al., 2020*), cells showed decreased transcriptional noise with aging. Of note, similar inconsistent results were found when using the pre-existing noise-measuring methods as compiled in *Decibel*, even when applying different methods to the exact same dataset (*Figure 2—figure supplement 2*). Overall, these results indicated that a generalized *increase in transcriptional noise* or a *loss of cellular identity* is not universal hallmark of aging, at least at the tissue level. However, the possibility that transcriptional noise increased in specific cell types was still unexplored by these analyses.

## The murine aging lung shows no consistent pattern of transcriptional noise at the cell type level and is instead characterized by reproducible alterations in immune cell composition

Do specific cell types become noisier as they age? In order to answer this question, we focused on a single tissue and conducted an in-depth analysis of transcriptional noise at the cell type level. For this, we selected the murine aging lung because of the relative abundance of available datasets in which

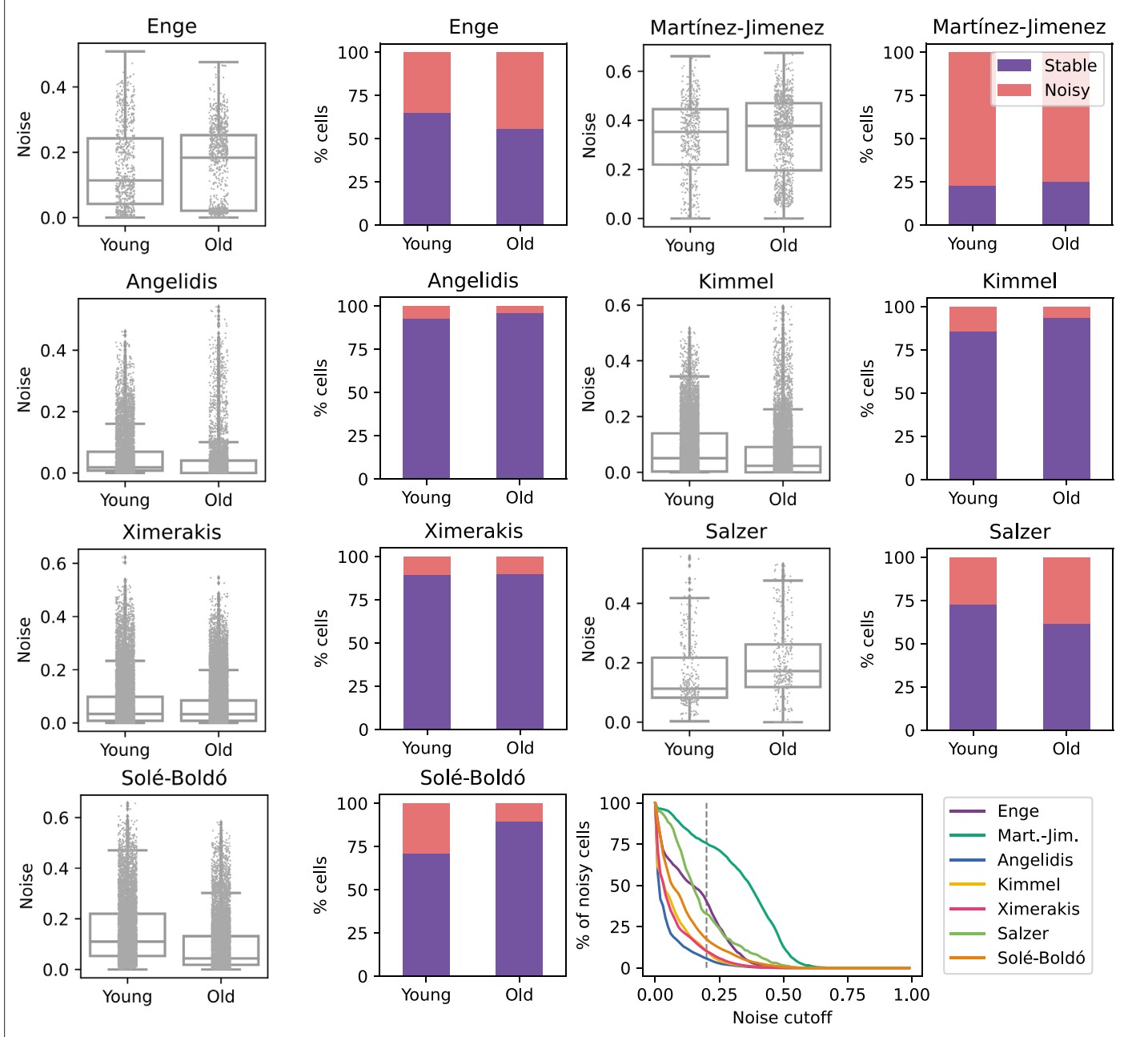

**Figure 2.** No consistent increase in transcriptional noise detected in seven single-cell RNA sequencing (scRNASeq) datasets of aging at the tissue level. The graphs show the amount of transcriptional noise, computed as 1 - membership to cell type clusters, in the young and old age groups of seven scRNAseq datasets of different tissues. For each dataset, the distribution of transcriptional noise values is shown as a stripplot over a boxplot, where the whiskers represent 1.5 times the interquartile range. Next to them the proportions of stable and noisy cells (noise ≥0.2) per age group are shown (purple bars = stable cells, pink bars = noisy cells). At the bottom right panel, the percentage of noisy cells with a transcriptional noise over a cutoff of 0.2 is plotted against the cutoff. Each colored line represents a different dataset.

The online version of this article includes the following figure supplement(s) for figure 2:

**Figure supplement 1.** Composition of the seven single-cell RNA sequencing datasets of aging used in the human aging lung analysis.

**Figure supplement 2.** Measurements of transcriptional noise on seven single-cell RNA sequencing datasets of aging using computational methods implemented in *Decibel*.

authors had reported an age-associated increase in transcriptional noise, yet restricted to particular cell types: *Angelidis et al., 2019*; *Kimmel et al., 2019*; and the Tabula Muris Senis (TMS) lung droplet and FACS datasets (*Almanzar et al., 2020*) (see *Appendix 5—figure 1*). In each dataset, transcriptional noise was measured as $1 - membership$ to cell type clusters in the young and old fractions, and the differences in median noise between the old and the young fraction for each of the existing 31 lung cell types and subtypes were calculated (*Figure 3*). Since changes in the gene expression of tissues can also be caused by altered cell type composition (*Trapnell, 2015*), we estimated the relative abundances of the 31 cell types in the young and old fraction of each dataset and measured the effect of age by fitting generalized linear models (GLM) to cell type composition data of the four datasets, using each mouse as a biological replicate (*Figure 3—figure supplement 1*). By plotting the age-related cell type enrichment together with the cell-to-cell transcriptional variability in each of the datasets, we obtained a comprehensive map of cell type enrichment and transcriptional noise associated with aging at the cell-identity level (*Figure 3A*). In this analysis, the direction and magnitude of changes in transcriptional noise varied across cell types. For instance, club cells (a bronchiolar exocrine cell type) were detected in sufficient numbers in three out of four datasets, their median membership score consistently decreasing 10–17% (which showed up as a moderate increase in transcriptional noise in *Figure 3A*; bubble #22). Similarly, lung interstitial fibroblasts' transcriptional noise appeared to increase with age, although with a larger range of membership scores (3–17%; bubble #24). In both cases, the cell type abundance was not affected by aging. In contrast, alveolar macrophages showed a decrease in age-associated transcriptional noise (5–12% increase in median membership; bubble #10). Finally, several cell identities appeared not to change significantly with regard to their transcriptional noise related to aging. That was clearly the case for capillary endothelial cells (bubble #9) and plasma cells (bubble #5). Vascular endothelial cells (bubble #6) showed less than 2% of change in noise in three out of four datasets but increased up to 8% in one dataset. Therefore, and contrary to expectation, quantitative analysis of age-associated transcriptional noise did not show a consistent pattern across diverse lung cell types in the four available datasets.

In contrast, the cell abundance analysis did reveal a strikingly consistent enrichment of immune cell types (lymphocytes in particular) across all datasets in old samples, indicative of immune cell infiltration in the old tissue. In particular, plasma cells (bubble #5) showed highly consistent enrichment in old mice, with an old/young odds ratio (OR) of 3 in the Kimmel dataset (p-value=1.1×10$^{-5}$) and of 9.3 in the Angelidis dataset (p-value=6.5×10$^{-21}$). The ORs for the two TMS datasets were most likely overestimated due to low cell numbers (only 9 and 22 old plasma cells were detected in the TMS datasets). The more abundant B cells (bubble #4) were also significantly enriched in 3/4 datasets (Angelidis: OR = 4.4, p-value=2.5×10$^{-69}$; Kimmel: OR = 1.2, p-value=6.3×10$^{-8}$; TMS FACS: OR = 2.0, p-value=8.9×10$^{-6}$). Other immune cell types such as monocytes, macrophages, and dendritic cells also appeared to be enriched in all datasets. This prompted us to further investigate the basis for the apparent immune cell enrichment and its potential relationship to increased transcriptional heterogeneity in the old age. In a qualitative approach to look for consistent patterns across datasets, we ranked cell identities according to their age-related increase in noise and enrichment (*Figure 3—figure supplement 2*). While most cell types were evenly distributed along the transcriptional noise ranking, this representation provided a visible distinction between immune and non-immune cell types regarding their age-related enrichment, with nearly all immune cell types appearing on top of the enrichment ranking. For instance, plasma cells (*Figure 3—figure supplement 2*, #5) were the third most enriched cell type in the Angelidis dataset and appeared on the top position in the rest of the datasets. Classical monocytes (#13) were found within the top 4 most enriched in 3/4 datasets. Interestingly, NK cells (#2) were the only underrepresented lymphocytes in old mice and ranked consistently in the least enriched positions among immune cells. Conversely, parenchymal cell types such as goblet cells, club cells, and ciliated cells consistently appeared at the bottom of the enrichment ranking, indicating that their proportion diminished with increased immune cell infiltration in the organ or, alternatively, loss of parenchymal cells associated with the old age. Endothelial cells were more evenly distributed along the ranking and thus did not show a clearly discernible age-associated enrichment or loss. Finally, we separated the lung cells into immune and non-immune cell categories and represented transcriptional noise and cell type enrichment values on a heatmap (*Figure 3B*). As clearly seen in this representation, the transcriptional noise increase associated with aging was extremely variable across cell identities and not always consistent across datasets (a comparison between these results and the

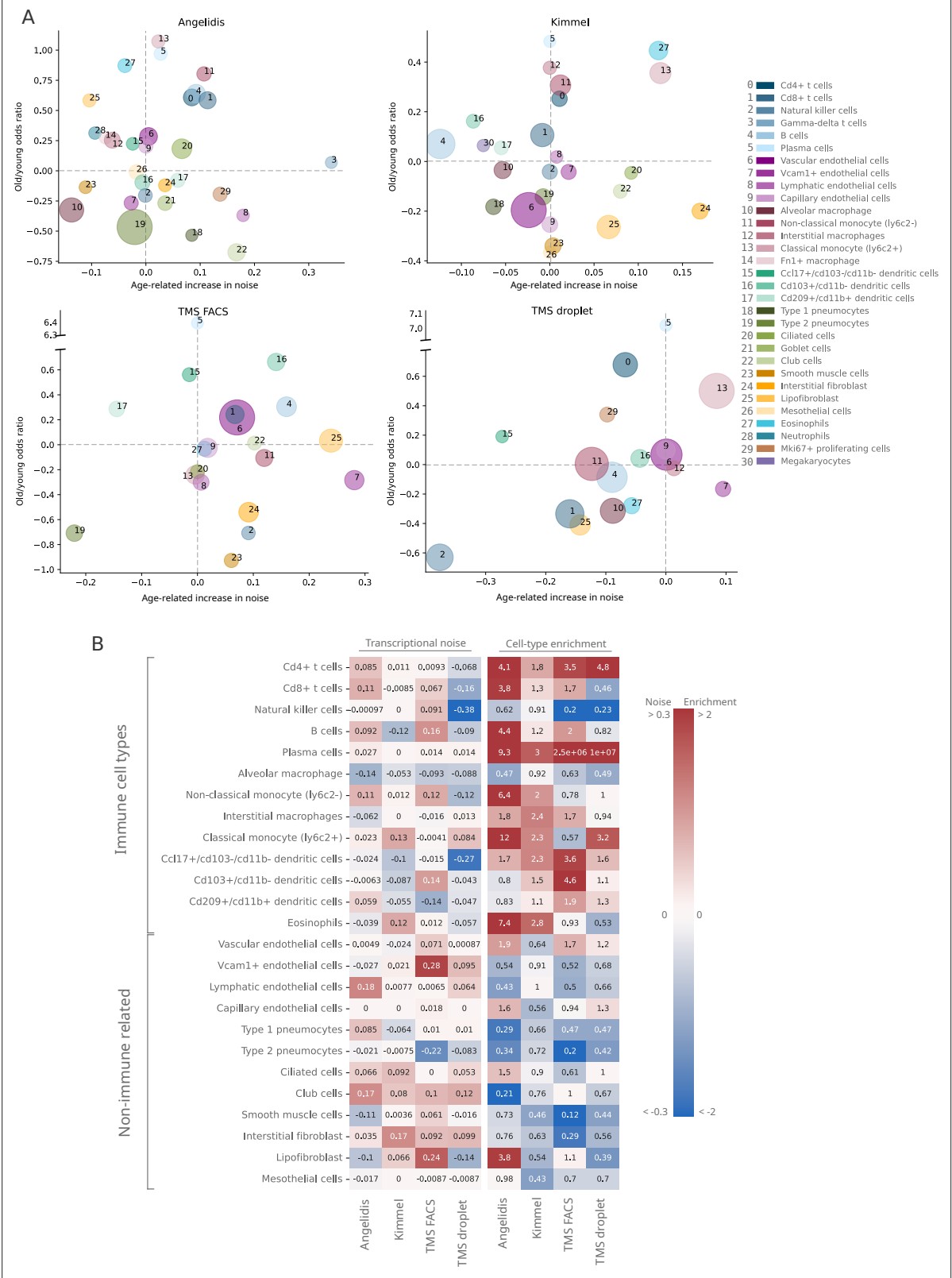

**Figure 3.** Lack of evidence for an increase in transcriptional noise of the murine aging lung and detection of an enrichment in immune cells. (**A**) Bubble chart of transcriptional noise and cell type enrichment (old/young odds ratio [OR]) of 31 murine lung cell identities. The age-related change in transcriptional noise (horizontal axis) is computed by *Scallop* as the decrease in median membership score per cell identity between young and old cells. The enrichment of each cell type in old samples with respect to their young counterpart is represented as the old/young OR (vertical axis). The

*Figure 3 continued on next page*

*Figure 3 continued*

area of the bubbles represents the expected proportion of each cell type in the whole dataset according to the binomial generalized linear model fitted for that dataset. (**B**) Immune cell type enrichment but not age-associated increase in transcriptional noise, is consistently detected in old mice lungs. The increase in transcriptional noise associated with aging (left) and the cell type enrichment (right) are shown for several lung cell identities classified on the left as immune and non-immune. Cell identities present in at least three out of the four studied datasets are shown, and the age-related difference in transcriptional noise of missing cell identities is imputed from the remaining three measurements (mean difference across datasets).

The online version of this article includes the following figure supplement(s) for figure 3:

**Figure supplement 1.** Composition of the four single-cell RNA sequencing datasets of the murine aging lung used in this figure.

**Figure supplement 2.** Qualitative ranking of murine aging lung cell types according to transcriptional noise and cell type enrichment.

**Figure supplement 3.** Comparison of the originally reported cell type-associated increase in transcriptional noise with the results obtained with *Scallop*.

results reported by *Angelidis et al., 2019* and *Kimmel et al., 2019* is provided in *Figure 3—figure supplement 3*). In contrast, the immune vs non-immune cell distinction alone explained the behavior of most cells with respect to their relative abundance with very few exceptions, namely NK cells and alveolar macrophages.

## Changes in the abundance of the immune and endothelial cell repertoires characterize the human aging lung

Our analysis of age-related cell type enrichment and increase in transcriptional noise in the murine lung highlighted the importance of the changes associated with the relative abundance of cell types that conform the aging lung. To test if this was specific of murine lungs or it could be a more generalized phenomenon, we conducted similar analyses on two large scRNAseq datasets of the aging human lungs (15,852 cells from 9 donors from the *Raredon et al., 2019* dataset and 15,048 cells from 2 donors of the human lung cell atlas (HLCA) dataset by *Travaglini et al., 2020*). We harmonized cell type labels between datasets by projecting the HLCA labels onto the Raredon dataset (*Figure 4—figure supplement 1*). Then, we calculated the difference in mean membership score between old and young cells for each cell type in the two datasets, together with the cell type enrichment using the GLM method as described earlier (*Figure 4*). In general, and similar to what we had previously observed in the murine aging lung, we found a lack of consistency between the two datasets regarding transcriptional noise associated with aging of specific lung cell types. However, we did observe some conserved changes in cell type composition. Particularly, many immune cell types were enriched in older donors, as in the murine aging lung. Plasma cells were significantly enriched in the Raredon dataset (OR = 2.6, p-values = 1.9e−6) and enriched, albeit not significantly, in the HLCA dataset (OR = 3.5e−11, p-value = 1). The latter result was most probably due to lack of statistical power, as the dataset only consists of two donors. Interestingly, alveolar macrophages were enriched (rather than depleted as in the murine aging lung) in both human aging lung datasets (OR = 1.2, p-value = 3.61e−5 in Raredon; OR = 2.8, p-value = 1.54e−261 in HLCA). Several endothelial cell types were significantly depleted in the two human aging lung datasets. Vein endothelial cells (OR = 0.65, p-value = 7.7e−5 in Raredon; OR = 0.58, p-value = 1.9e−9 in HLCA), capillary endothelial cells (OR = 0.81, p-value = 3.4e−2 in Raredon; OR = 0.3, p-value = 3.5e−141 in HLCA), endothelial cells of lymphatic vessels (OR = 0.51, p-value = 1.9e−9 in HLCA). These results indicated that aged human lungs present reduced vascularization and significant immune cell infiltrates as compared to the young. Even though the evidence for changes in tissue composition is based on a single tissue, we hypothesize that these facts may have influenced previous analyses of transcriptional noise associated with aging.

## Distance-to-centroid methods detect transcriptionally stable cell subtypes as transcriptional noise

A relevant open question is what was the source of apparent transcriptional *noise* in previous studies that were based on DTC methods. Since we found important changes in the community of human alveolar macrophages in the HLCA dataset, we conducted an in-depth analysis on that cell type that revealed four distinct alveolar macrophage communities that emerge with aging from a single transcriptionally homogeneous cluster (see *Figure 5—figure supplement 1*). The four aged alveolar macrophage subclusters present a markedly different expression of genes coding for surfactant proteins (*SFTPA1*, *SFTPA2*, *SFTPB*, *SFTPC*, and *SFTPD*), i.e., they show changes consistent with

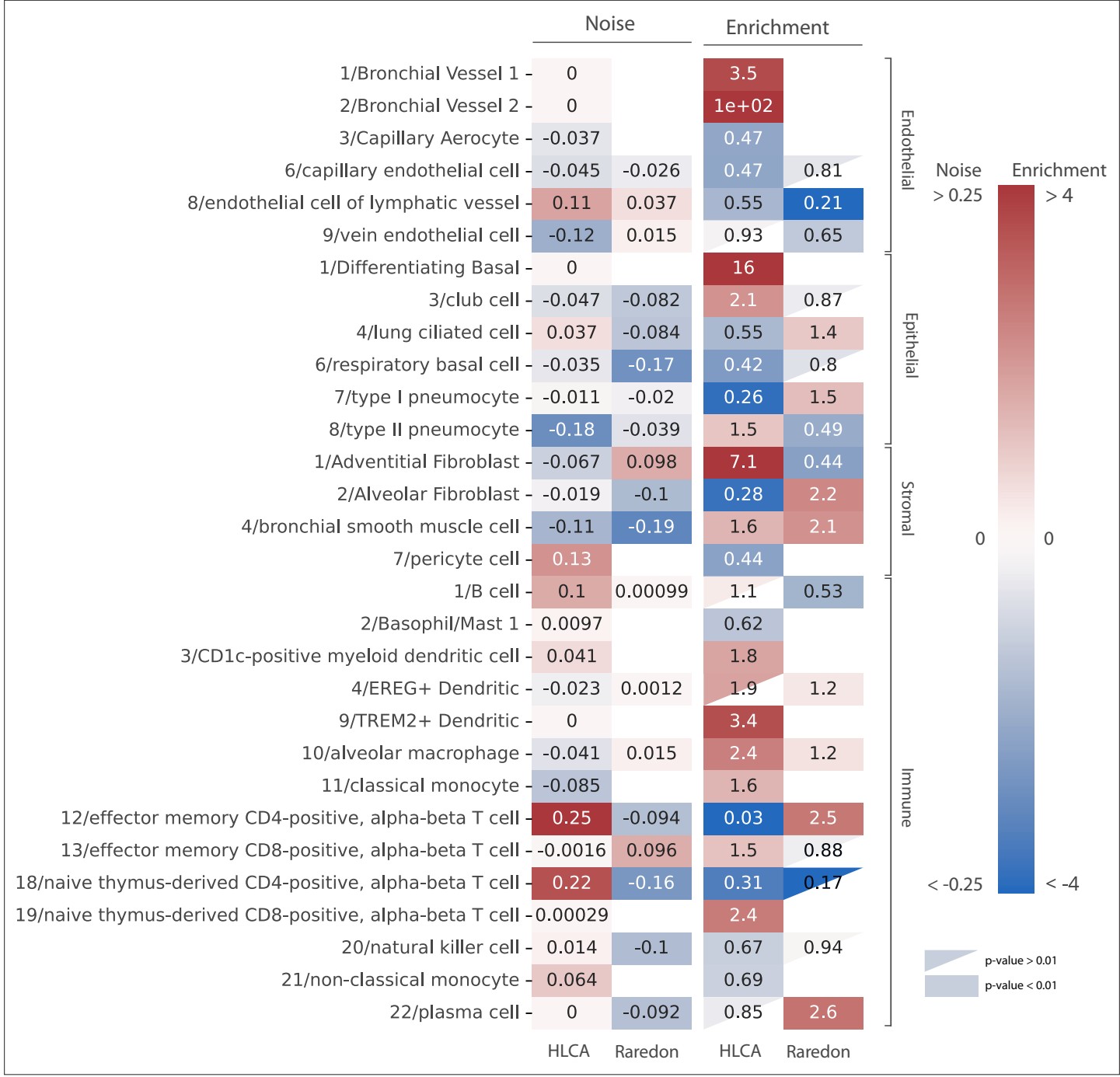

**Figure 4.** Human aging lungs show no increase in transcriptional noise but consistent depletion and enrichment of specific endothelial and immune cell types. The increase in transcriptional noise associated with aging (*Noise*, left) and the cell type enrichment (*Enrichment*, right) values are shown for 30 human lung cell identities as detected in the human lung cell atlas (HLCA) and Raredon datasets (*Raredon et al., 2019*; *Travaglini et al., 2020*). For each cell type, its age-related increase in noise (difference in $1 - membership$ between old and young individuals per cell type) and the old/young odds ratio (OR) are shown. Only cell types whose enrichment/depletion is statistically significant in at least one of the datasets are shown, and the ORs associated with a p-value >0.01 are shown as a triangle. The color-bar for the enrichment is shown in a logarithmic scale.

The online version of this article includes the following figure supplement(s) for figure 4:

**Figure supplement 1.** Composition of the two single-cell RNA sequencing datasets of the human aging lung used in this figure.

alternative fate determination. Deregulated surfactant protein expression is connected to the age-related functional decline of human lungs. In fact, mutations in the gene coding for surfactant protein C (*SFTPC*) and in the *MUC5B* promoter region are linked to pulmonary fibrosis, but the effects of these mutations are usually not observed until late in life (around 60–70 years old), because age-related decline in proteostasis is needed for aggregation prone or misfolded proteins to actually cause damage (*Schneider et al., 2021*). Interestingly, we measured the age-associated transcriptional noise in the alveolar macrophage community using a DTC method (*Euclidean distance to the cell type mean*) and *Scallop*, and observed that only the latter algorithm could accurately detect the

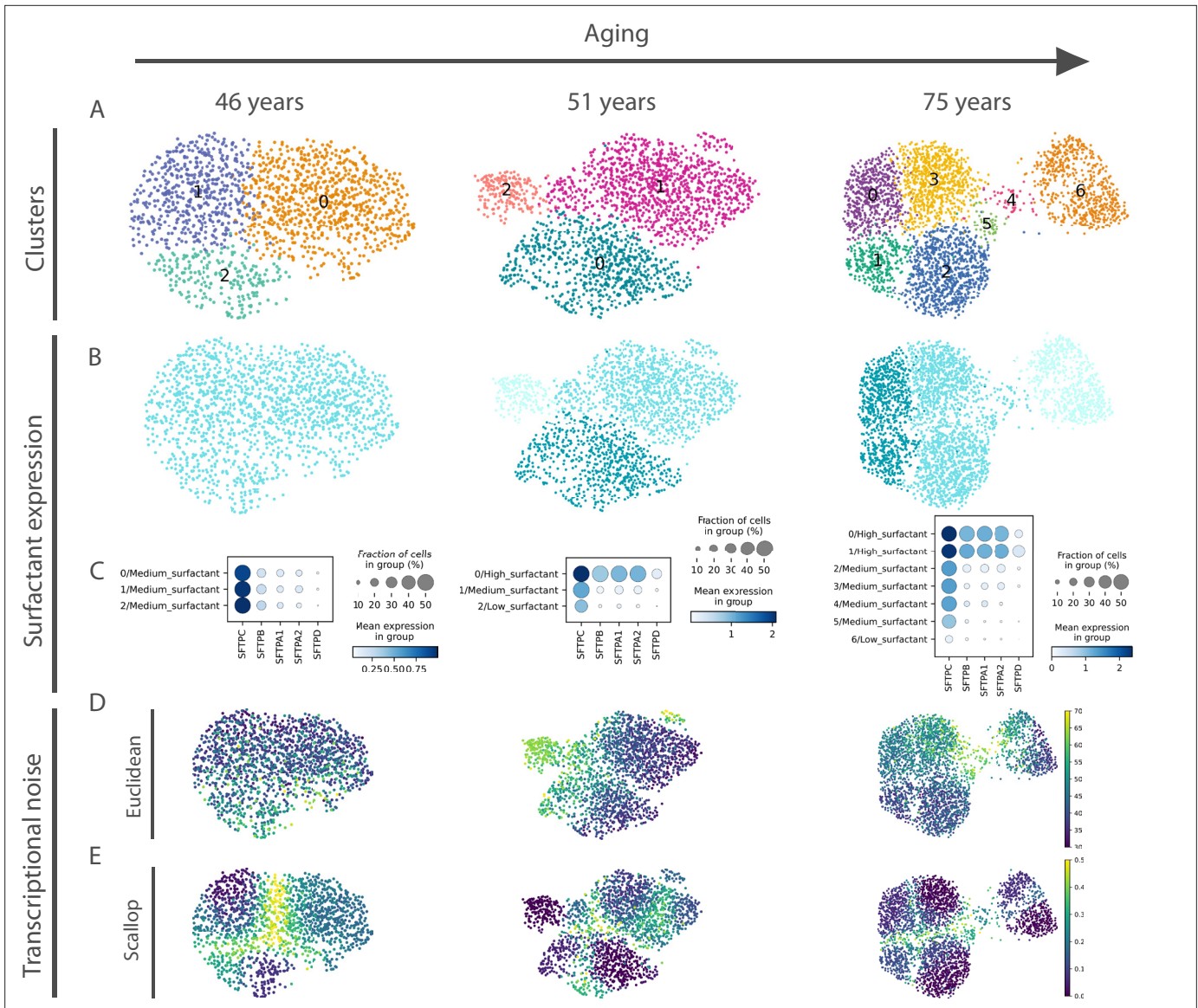

**Figure 5.** Euclidean distance-to-centroid methods are unable to distinguish *bona fide* transcriptional noise from alternative cell fate specification. (**A**) An increasing number of alveolar macrophage subclusters (as obtained with Leiden) are detected in three donors (aged 46, 51, and 75 years) from the *Travaglini et al., 2020* (human lung cell atlas) dataset. (**B–C**) The new cell clusters are characterized by differential surfactant protein gene expression levels, as clearly seen on the uniform manifold approximation and projection (**B**) and dotplot (**C**) representations. (**D–E**) Transcriptional noise measurements, using the Euclidean distance to cell type mean (**D**) and *1 − membership* using *Scallop* (**E**), demonstrate that only the latter method is able to distinguish *bona fide* transcriptional noise from the formation of new clusters that are transcriptionally stable.

The online version of this article includes the following figure supplement(s) for figure 5:

**Figure supplement 1.** Expression of surfactant protein genes by human alveolar macrophages.

**Figure supplement 2.** Alveolar macrophages are the most affected by aging.

emergence of distinct and transcriptionally stable alveolar macrophage subpopulations, whereas the DTC method would interpret this specification of alternative cell fates as a single macrophage population undergoing loss of identity (*Figure 5*).

Additionally, a deeper analysis of the pancreatic β-cells from the human aging pancreas dataset (*Enge et al., 2017*) revealed that the transcriptional stability attributed to young cells might in fact be a consequence of the distinct transcriptional profile of the cells from one of the two donors that constitute the young cohort (see *Figure 5—figure supplement 2*). According to some methods (*ERCC-based* and *euc_dist_invar*), the cells from the 21-year-old patient present lower transcriptional noise than the rest of the patients. But this lower transcriptional noise appears to be restricted to one donor (the 21-year-old donor). In fact, the other young donor (22 years old) presents transcriptional noise values that are similar to those of the old-cohort donors (38–54 years old). Moreover, the insulin expression level of the 21-year-old donor seems to be substantially different to the rest of the donors. We hypothesize that *ERCC-based* and *DTC* methods yielded low transcriptional noise measurements for those cells due to their distinct transcriptional profile (possibly unrelated to aging), which in turn resulted in a significant difference between the young and the old cohorts regarding transcriptional noise. We show that *Scallop* did not detect noise associated with aging for this set of pancreatic *beta*-cells.

## Discussion

Mechanistically, it is not clear if aging is a tightly regulated process or may be the result of passive phenomena of stochastic nature (*Kirkwood and Melov, 2011*; *Schmeer et al., 2019*; *Gladyshev, 2016*; *da Costa et al., 2016*). In the absence of further mechanistic insight, aging is characterized by a series of phenotypic changes at the cellular and tissue levels, such as genomic instability, epigenetic alterations, chronic low level inflammation (*inflammaging*), immunosenescence, and impaired regeneration (*López-Otín et al., 2013*; *Gems and de Magalhães, 2021*). In addition, an increase in transcriptional noise has been observed in some aged tissues and cell types (*Enge et al., 2017*; *Martinez-Jimenez et al., 2019*; *Angelidis et al., 2019*; *Kimmel et al., 2019*). Transcriptional noise could be related to genomic instability (*Vijg, 2021*), epigenetic deregulation (*Lu et al., 2020*; *Oliviero et al., 2022*), or loss of proteostasis (*Li et al., 2020*), all established hallmarks of aging. Some authors consider transcriptional noise to be a hallmark of aging in and of itself (*Mendenhall et al., 2021*).

In any case, the origin of transcriptional noise is unclear, as it could arise from many different sources. Most importantly, it is not possible to distinguish between intrinsic and extrinsic noise from a snapshot of cellular states, i.e., one cannot tell whether the observed differences between cells in a single-cell RNA experiment reflect time-dependent variations in gene expression or differences between cells across a population (*Ham et al., 2021*). Interestingly, recent work by *Liu et al., 2017* measuring intrinsic noise in *S. cerevisiae* showed that aging is associated with a steady decrease in noise, with a sudden increase in soon-to-die cells. They proposed that noise decreasing in normal aging was due to increased rates in chromatin state transitions, and showed that disrupting chromatin structure at the gene promoter was sufficient to change this course. Another longitudinal study found an increase in extrinsic noise and a lack of change in intrinsic noise in diploid yeast (*Sarnoski et al., 2018*).

Since aging is multifactorial, and mutational load most likely leads to clonal expansion of aberrant cells that accumulate throughout the lifetime of the individual, other authors suggest that aging traits may be associated with cell type imbalance in aged organs (*Cagan et al., 2022*). Another recent hypothesis is *inter-tissue convergence* through age-associated loss of specialization (*Izgi et al., 2022*). Our results suggest that transcriptional noise is not a *bona fide* hallmark of aging. Instead, we posit that previous analyses of noise in aging scRNAseq datasets have been confounded by a number of factors, including both computational methods used for analysis as well as other biology-driven sources of variability.

In this work, we made a systematic comparison of the most important families of methods that have been used to quantify age-related transcriptional noise through the implementation of *Decibel*, a novel Python toolkit. Since we were not convinced of the utility of these methods to determine *bona fide* transcriptional noise, we developed a novel method and applied it to a wide array of tissues. Our proposed tool, *Scallop*, presents some advantages over existing methods: it does not require neither ERCC spike-ins nor cell type labels. In addition, it provides information

that is complementary to the GCL, as it yields a cell-wise measurement of transcriptional noise that enables us to compare between *stable* and *unstable* cells within the same cluster or cell type. Most importantly, *Scallop* measures transcriptional noise by membership to cell type-specific clusters which is a re-definition of the original formulation of *noise* by Raser and O'Shea: measurable variation among cells that should share the same transcriptome. This is in stark contrast to measurements of *noise* including other phenomena (as demonstrated in *Figure 5*) by the DTC methods prevalent in the literature. Interestingly, *Scallop* appears to detect transcriptional noise driven by the lack of expression of cell type markers in a very robust way, and this should make it a better choice for measuring loss of cell type identity compared to DTC methods. When applied to seven independent aging datasets, the results obtained revealed little overlap in the magnitude and directionality of the changes in transcriptional noise associated with aging of the different tissues analyzed, providing evidence that an increase in transcriptional noise might not be as evident as generally thought.

In order to investigate cell type-specific effects in transcriptional noise, it is crucial to compare between different datasets of the same aging tissue. Otherwise, it is difficult to ascertain whether the variability observed between cell types is due to a pattern that is conserved in that tissue or is merely the effect of the intrinsic variability associated with scRNAseq experiments (*Fonseca Costa et al., 2020*). For the cell type-specific study, we focused on the aging lung, as the effect of aging of this tissue has gained relevance (*Schiller et al., 2019*) due to its association with chronic obstructive pulmonary disease, lung cancer, and interstitial lung disease (*Angelidis et al., 2019*; *Schneider et al., 2021*) and its increased risk of severe illness in COVID-19 patients (*Williamson et al., 2020*). In the 31 cell types analyzed in mouse lungs, we found increased transcriptional noise in club cells and interstitial fibroblasts only, while alveolar macrophages seemed to decrease it. Of interest, a single-cell analysis of alveolar macrophages did not identify distinct clusters associated with mouse or human aging, identifying changes in the aged alveolar microenvironment as key for their altered functionalities (*McQuattie-Pimentel et al., 2021*). In humans, we analyzed two aging lung datasets that provide complementary information, as the final Raredon dataset consists of 9 donors of a wider range of ages but is not as well powered in terms of cell type resolution as the HLCA dataset, which contains 48 cell identities. Similar to what we had previously observed in the murine aging lung, there was no consistency between the 2 datasets regarding transcriptional noise of the 30 specific cell types detected. However, both in human and mouse lungs we detected a shift in the abundance of a number of cell populations with age, most clearly seen for immune cells.

In fact, the age-associated increase in immune cell infiltration of solid organs may be generalized. Specifically, one study found neutrophil and plasma cell infiltration in adipose tissue, aorta, liver, and kidneys of aged rats of both sexes, and the immune cell infiltration was reversed by caloric restriction (*Ma et al., 2020a*). Another study found a subtype of highly secretory plasma cells infiltrated in the aged bone marrow, spleen, fat, kidney, heart, liver, muscle, and lungs (*Schaum et al., 2020*). Of note, immune cell senescence has been shown to induce aging of solid organs (*Yousefzadeh et al., 2021*), in what has been proposed to be a feed-forward circuit (*Salminen, 2021*). Therefore, the importance of immune cell infiltration of the aged lungs cannot be overlooked. In fact, age-associated immune cell type enrichment has also been observed in two independent studies of macaque lungs. One study found increased mast cells, plasma cells, and CD8+ T cells in aged lung tissue (*Ma et al., 2020b*), while the other found increased alveolar and interstitial macrophage numbers in bronchoalveolar lavages of old macaques (*Rhoades et al., 2022*). The significance of the shift in cellular composition of the aged lungs in relation to the appearance of aging traits remains to be determined. Of note, alternative explanations for transcriptional changes associated with aging such as *tissue convergence* are compatible with shifts in the cellular composition of aging tissues and organs being a primary cause of convergence (*Izgi et al., 2022*).

In summary, the sources of the apparent increase in *transcriptional noise* reported by previous studies may be multiple and are mostly related to the computational methods used to characterize transcriptional noise and cellular identity in aged tissues. Open availability of *Decibel* and *Scallop* represents an opportunity for the aging research community to further investigate these issues, and they are also valuable for researchers addressing cell-to-cell variability of scRNAseq datasets in other settings.

# Methods

## *Decibel*: Python toolkit for the quantification of transcriptional noise

We developed a Python toolkit for the quantification of loss of cell type identity associated with aging. We implemented methods as they were originally described in the literature.

### Biological variation over technical variation

Biological variation over technical variation is measured as in the original formulation by *Enge et al., 2017* by computing $1 - \rho$, where $\rho$ is the Pearson's correlation between the gene expression vector of each cell and the mean expression of its cell type, i.e., the gene expression averaged over all the cells from the same cell type and individual. For each cell type and individual mouse or donor, the mean gene expression vector – the averaged expression of the whole set of monitored genes across cells – is computed. Then, the biological variation is measured as the Euclidean distance from each cell to its cell type mean for that individual. The technical variation is computed using the same procedure but using only the ERCC spike-ins in the calculation of the distance to cell type mean. Finally, the transcriptional noise is calculated by dividing the biological variation by the technical variation per cell.

### Euclidean distance to cell type mean

The distance to the cell type mean is measured as the second method described by *Enge et al., 2017*. For each cell type and individual mouse or donor, we compute the average whole-transcriptome expression. The noise is quantified as the Euclidean distance between the gene expression vector of each cell and its corresponding individual-matched cell type mean expression vector.

### Invariant gene-based Euclidean distance to tissue mean

This is the third method described by *Enge et al., 2017*. It is computed as the Euclidean distance from each cell to the average expression across cell types using a pre-selected set of invariant genes that is selected as follows: first, genes are sorted according to their mean expression and split into 10 equally sized bins, and the 2 extreme bins are discarded (10% most expressed and 10% least expressed genes). Then, the 10% of genes with the lowest coefficient of variation within each bin are selected and used for the calculation of the Euclidean distance between the mean expression vector across cell types and each of the cell expression vectors.

### Average global coordination level

Taking the Matlab code provided by the authors, we implemented the GCL in Python. As the original formulation was used in datasets with a single cell type, here we computed the GCL for each cell type separately and then calculated the average GCL for the tissue. For each cell type, the GCL was calculated by splitting the whole transcriptome into two random halves and computing the batch-corrected distance correlation between them (*Levy et al., 2020*). The GCL per cell type was averaged over k times. Following the authors' recommendation, we used k=50 in all of our calculations.

## *Scallop*

*Scallop* iteratively runs a clustering algorithm of choice (default: Leiden; *Traag et al., 2019*) on randomly selected subsets of cells. Then, it computes the frequency with which each cell is assigned to the most frequently assigned cluster. *Scallop* has three key steps: (1) bootstrapping, (2) mapping between cluster labels across bootstrap iterations, and (3) computation of the membership score.

Leiden is a graph-based community detection algorithm that was designed to improve the popular Louvain method (*Blondel et al., 2008*). Graph-community detection methods take a graph representation of a dataset. In the context of single-cell RNAseq data, shared nearest neighbor graphs are commonly used. These are graphs whose nodes represent individual cells, and edges connect pairs of cells that are part of the K-nearest neighbors of each other by some distance metric. The aim of community detection algorithms like Leiden is to find groups of nodes that are densely connected between them, by optimizing modularity. For a graph with C communities, the modularity (Q) is computed by taking, for each community (group of cells), the difference between the actual number of edges in that community ($e_i$) and the number of expected edges in that community ($\frac{K_i^2}{2m}$):

$$Q = \sum_{i=1}^{C} \left\{ e_i - r \cdot \frac{K_i^2}{2m} \right\}$$

where $r$ is a resolution parameter ($r>0$) that controls for the amount of communities: a greater resolution parameter gives more communities whereas a low-resolution parameter fewer clusters. Since maximizing the modularity of a graph is an NP (non-deterministic polynomial time)-hard problem, different heuristics are used, and Leiden has shown to outperform Louvain in this task both in terms of quality and speed (*Traag et al., 2019*). However, users can choose to run the *Louvain* method instead by setting the parameter clustering = louvain in the initialization of the Bootstrap object.

## Bootstrapping

*Scallop* runs a community-detection algorithm on subsets of cells drawn from the original dataset. The subsets are selected randomly with replacement from the whole population (the seed can be defined by the user). The number of cells to be selected on each bootstrap iteration is computed through the fraction of cells user-defined parameter `frac_cells` (default: 0.95). The community detection algorithm is run `n_trials` times (default: 30). An additional clustering is run with all the cells (`frac_cells = 1`) for it to be used as a reference in the mapping stage. A bootstrap matrix (`n_cells×n_trials`) is obtained that contains the cluster labels that have been assigned to each cell on each bootstrap iteration. The cluster labels are the ones obtained from the python implementation of Leiden through the *Scanpy* function `sc.tl.leiden()` and are numbered from '0' to $n$ according to the size of the cluster, i.e., the cluster with the highest number of cells is assigned the label '0', the second most abundant is assigned the label '1', and so on. Since the subset of cells used in each run is different, clustering results vary from run to run, and labels are not comparable between bootstrap iterations.

## Cluster relabeling

In order to compare between cluster assignments from different bootstrap iterations, cluster identities need to be relabeled. A contingency table is computed between each clustering solution in the bootstrap matrix and a reference clustering, which was obtained by running the community detection algorithm on all the cells. From the original bootstrap matrix, we obtain a relabeled bootstrap matrix. The assumption is made that if cluster A from bootstrap iteration $i$ and cluster B from bootstrap iteration $j$ have a large number of cells in common, then they should have the same label. In order to find the mapping between clusters, an overlap score matrix is computed for every column in the bootstrap matrix against the reference labels. The overlap score ($S$) between cluster A from the reference clustering solution ($A_{ref}$) and cluster B from the $i$-th iteration ($B_i$) is defined as follows:

$$S(A_{ref}, B_i) = \frac{|A_{ref} \cap B_i|}{|A_{ref}|} + \frac{|A_{ref} \cap B_i|}{|B_i|}$$

where $|A_{ref}|$ and $|B_i|$ are the number of cells in the cluster $A$ and $B$ from the reference clustering solution and the $i$-th bootstrap iteration, respectively, and $|A_{ref} \cap B_i|$ is the number of cells in common between the two clusters. The score is then [0–1]-scaled by dividing it by the maximum score: 2. The maximum score would correspond to a total overlap between the two clusters.

The score is computed for every pair of clusters between the reference solution and each of the bootstrap iterations to obtain a contingency matrix ($n\_clusters_{ref} \times n\_clusters_i$). In order to find the optimal mapping between the two clustering solutions, we search for the permutation of the columns that maximizes the trace of the contingency matrix. We do this by using Munkres, a Python implementation of the Hungarian method (*Munkres, 1957*).

As the reference clustering solution is computed on the whole dataset but each of the bootstrap iterations is run on a subset of cells (*frac_cells*), the number of clusters obtained in each iteration might not be equal to the number of clusters in the reference. In order to deal with this, we consider three cases:

1. The number of clusters in the reference clustering solution is equal to the number of clusters obtained in the $i$-th bootstrap iteration. This case is dealt with easily, as the Hungarian method yields a 1:1 mapping between the two clustering solutions.

2. Fewer clusters are obtained in the *i*-th bootstrap iteration than in the reference solution. This may happen if one or more clusters from the reference are merged into a single cluster in a bootstrap iteration. In this case, a 1:1 mapping is obtained, but one or more of the cluster labels from the reference clustering remain unused.

3. More clusters are obtained in the *i*-th bootstrap iteration than in the reference solution. This may happen if one cluster from the reference is further divided into two or more subclusters in a bootstrap iteration. A 1:1 mapping is obtained, but one or more clusters from the bootstrap iteration remain unmapped. Usually, this means that a cluster from the reference solution was divided into two or more subclusters in the bootstrap iteration. In this case, the subcluster with the largest overlap degree with one of the clusters in the reference clustering solution receives its label. The other subcluster remains unmapped. When this happens, those clusters are flagged as *unmapped*. Then, an additional mapping step is carried out between the *unmapped* clusters from all bootstrap iterations. This is done by creating an overlap score matrix similar to the one created in the mapping process and searching for the permutation of the columns that maximizes its trace, using the Hungarian method. In order to avoid spurious mappings between unrelated *unmapped* clusters, a minimum overlap score of 0.1 is imposed for two *unmapped* clusters to be renamed as the same cluster.

## Computation of the membership score

*Scallop* computes three different membership scores: a frequency score ('freq'), an entropy score ('entropy'), and a Kullback-Leibler ('KL') divergence score. We use the frequency score here as it yields results that are consistent to the ones obtained with the other two alternative scores, and its meaning is more intuitive than those of the two alternative methods. The frequency score is computed as the fraction of bootstrap iterations where a cell was assigned to the most frequently assigned cluster label. In order for the score to take values between 0 and 1, only the cells selected in each bootstrap run are considered as the total number of cells. More information on the calculation of the entropy and the KL scores can be found in the *Scallop* documentation.

$$Freq\,score(c) = \max \left\{ \frac{|c_n|}{\sum_{m \in clusters} |c_m|} \mid n \in clusters \right\}$$

where $|c_n|$ is the number of times cell $c$ was assigned to the *n*-th cluster, and $\sum_{m \in clusters} |c_m|$ is the sum of all assignments made on cell $c$, which is the same as the number of times cell $c$ was clustered across bootstrap iterations.

## Validation of *Scallop*

### Performance of *Scallop* and two DTC methods on four artificial datasets with increasing transcriptional noise

We used four artificially generated datasets with various degrees of transcriptional noise (*Figure 1—figure supplement 1*). Each of the four datasets consists of 10 K cells, from nine populations (named *Group1-Group9*) with the following relative abundances: 25, 20, 15, 10, 10, 7, 5.5, 4, and 3.5%. The four datasets only differ in the de.prob parameter used in their generation. The de.prob parameter determines the probability that a gene is differentially expressed between subpopulations within the dataset. The greater the de.prob value, the more DEGs there will be between clusters, meaning that the different cell types present in the dataset will cluster in a more robust way. Decreasing the value of de.prob results in datasets with noisy cells, with populations that do not have such a strong transcriptional signature. In order to study how *Scallop* can capture the degree of robustness with which cells of the same cell type cluster together, we selected four de.prob values and obtained four datasets that represent *low*, *medium low*, *medium high*, and *high* noise levels. We then measured transcriptional noise using *Scallop* and two alternative DTC methods implemented in Decibel: (1) whole transcriptome-based Euclidean distance to cell type mean and (2) invariant gene-based Euclidean distance to tissue mean expression. GCL measurements were not carried out here as the method does not yield a transcriptional noise measurement per cell, so no comparisons can be made with respect to the amount and localization of noisy cells the method is able to detect within a cluster. Also, computing the ERCC spike in-based transcriptional noise was not possible for artificial datasets.

## Ability to detect noisy cells within cell types

Using three of the four datasets used in the previous section (the ones corresponding to *medium low*, *medium high,* and *high* noise levels), we plotted the top 10% noisiest (lowest membership) and the top 10% most stable (highest membership) cells (see *Figure 1—figure supplement 2*).

## Effect of cellular composition

We simulated five artificial datasets with the same nine cell type populations and the same total number of cells, but whose relative abundances were different between them. We used the *ID* (*Ortigosa-Hernández et al., 2017*) to measure class imbalance in each of them and to make sure that the selected cell compositions represented a wide range of IDs (to this end, we explored ID values between 1.2 and 5.3). The ID provides a normalized summary of the extent of class imbalance in a dataset in so-called 'multiclass' settings (where more than two classes are present). It was specifically developed to improve the commonly used imbalance ratio measurement, where only the abundance of the most and the least popular classes is considered in the calculation. We used the python implementation by *Juez-Gil, 2021* of the *ID*. The most imbalanced dataset (ID = 5.3) was generated by randomly subsampling 4500 cells from the original *medium high noise* 10 K-cell artificial dataset (generated with *Splatter* using de.prob=0.001). The rest of the datasets was generated by randomly selecting a known number cells from each cell type. The number of cells per cell type and the percentage they represent are shown in *Figure 1—figure supplement 3B*.

## Effect of dataset size

The effect of dataset size (total number of cells) on the transcriptional noise measured by *Scallop* was tested by generating differently sized versions of an artificial dataset (see *Figure 1—figure supplement 4*). We did this by randomly subsampling cells without replacement from the 10 K *medium high* noise dataset generated with *Splatter* (de.prob=0.001). We created 10 datasets sized 1000–10,000 cells. Each of the datasets was processed again after subsampling (highly variable gene detection, principal component analysis (PCA), neighbor calculation, and UMAP) prior to noise calculation.

## Effect of feature expression

We evaluated the effect of the number of genes on transcriptional noise by generating 10 datasets with a number of genes between 5000 and 14,000 (see *Figure 1—figure supplement 5*). We did this by subsampling genes without replacement from the 10 K *medium high* noise dataset generated with *Splatter* (de.prob=0.001). Datasets were processed again after subsampling before running *Scallop*.

## Effect of cell type marker expression

In order to measure the effect, gene marker expression has on the membership with which cells are assigned to their cell type cluster, we ran a simulation where the top 10 markers for a cell type were removed from the dataset one by one, and the cumulative effect of removing them was measured (see *Figure 1—figure supplement 6*). We selected the most stable cell type from the 10 K *medium high* noise dataset generated with *Splatter* (de.prob=0.001) so that the first simulation lacked the expression of the *Top1* marker, the second simulation had the effect of the first two markers removed (*Top1* and *Top2*), and so on. Then, we ran *Scallop* on each of the resulting datasets and observed a steady increase in transcriptional noise associated with that cell type.

## Ability to detect stable and unstable cells in the 8K human T cells

We downloaded a 23,766 PBMC dataset from from 10× Genomics. We ran the standard processing pipeline including highly variable gene detection, dimensionality reduction through PCA and UMAP, and clustering. We annotated the dataset according to PBMC marker expression and selected the cluster of T lymphocytes (see *Figure 1—figure supplement 7*). We obtained a dataset of 8278 cells. We ran the processing pipeline on the T lymphocyte dataset and obtained three main clusters of cells, which we annotated as *0/CD4+ T* cells, *1/CD4+ T* cells, and *2/CD8+ T* cells according to their expression of the gene markers *CD3C, CD3D, CD3E, CD4, CD8A,* and *CD8B*. Then, we calculated the whole transcriptome-based Euclidean distance to the cell subtype mean (*euc_dist*), the invariant gene-based Euclidean distance to the T cell mean (*euc_dist_tissue_invar*), the *Scallop* noise as 1-membership

(*scallop_noise*), and the GCL per T subtype. We selected the 10% most stable and 10% most unstable cells as those having the lowest and highest noise scores according to two methods: *euc_dist* and *scallop_noise*.

## Robustness to input parameters

We selected a set of five scRNAseq datasets of various sizes and depths (Table, Lack of evidence for increased transcriptional noise in aged tissues). Three datasets were taken from published scRNAseq studies (*Paul et al., 2015*; *Moignard et al., 2015*; *Joost et al., 2016*), and two were from 10× Genomics (PBMC3K, Heart10K). We computed the membership scores of all the cells in 5 datasets 100 times, on a range of bootstrap iterations (n_trials), fraction of cells used in each iteration bootstrap (frac_cells) and resolution (res) values. We then computed the median correlation distance between the 100 runs of *Scallop* with each set of parameters (see *Figure 1—figure supplement 8*). We used the spatial.distance.correlation method from Scipy to compute the correlation distance.

## Statistical significance of differential expression of PBMC markers

The assessment of the cell-to-cell variability associated with aging using *Scallop* relies on the assumption that cell-to-cell variability is caused by transcriptional noise, and that it can be measured by evaluating cluster stability. We checked our assumption by comparing the transcriptomic profiles of the cells that had a high and a low membership score (measured using *Scallop*) (see *Figure 1—figure supplement 9*). *Stable* cells should have a more robust expression of cell type markers than the *unstable* cells. We downloaded the PBMC 3K dataset from 10× Genomics. After running the standard processing pipeline, we ran *Scallop* on the dataset and selected the most stable and most unstable half of the cells within each annotated cluster. For each cell type, we defined the most stable cells as those with a membership score greater than the median membership score of that cell type. Hence, we compared two sets of cells (stable vs unstable) of the same size, and we analyzed the effect size and statistical significance of a routine downstream analysis (differential expression) when given each of the sets as input. We computed the 100 most DEGs between each cell type and the rest of the cells using (1) all cells, (2) only the stable cells, and (3) only the unstable cells. B cells and megakaryoctes were excluded from the analysis as the former was highly stable (so we could not compare between the stable and the unstable fraction) and the latter consisted of very few cells. We compared the distribution of log-fold changes and p-values associated with those DEGs when using only the stable, only the unstable, and all the cells.

## Single-cell RNA sequencing data processing

### 2,5K human aging pancreatic cells

The raw count matrices and the metadata files from *Enge et al., 2017* were downloaded from the Gene Expression Omnibus (accession number: GSE81547). The separate GSM (GEO sample accession) files were merged into a single raw count matrix and processed them using the following pipeline in Scanpy (*Wolf et al., 2018*): filtering of low quality cells and genes, normalization, log-transformation of counts, PCA, batch-effect correction using harmony (*Korsunsky et al., 2019*), Leiden community detection (resolution = 1.0), and UMAP dimensionality reduction. 11 clusters were obtained and annotated using the expression of the markers *INS* (β cells), *GCG* ($\alpha$ cells), *SST* ($\delta$ cells), *PRSS1* (acinar cells), *PROM1* (ductal cells), *PPY* (PP cells), and *THY* (mesenchymal cells). Donors were classified into three categories as in the original work by *Enge et al., 2017*: 'pediatric' (0–6 years old), 'young' (21–22 years old), and 'old' (38–54 years old). Samples from pediatric donors were not used in the aging analysis (see Inclusion criteria, Lack of evidence for increased transcriptional noise in aged tissues).

The analysis of the pancreatic β-cells (*Figure 5—figure supplement 2*) was done by selecting β-cells from the pancreatic aging cell dataset and re-running the highly variable gene detection, PCA, neighbor computation, and UMAP steps.

## 1,5K murine aging CD4+ T cells

We downloaded the raw data and metadata files from *Martinez-Jimenez et al., 2019* from the authors' GitHub. We created an annData object with the raw count matrix and the metadata (mouse strain, age-group, stimulus, individual, and cell type). We identified and flagged the counts corresponding to ERCC spike-in controls. We ran a standard processing pipeline: filtering out low-quality cells and genes, normalization and log-transformation of counts, selection of highly variable genes, batch-effect correction between mouse strains (*Mus musculus domesticus* and *Mus musculus castaneus*), and dimensionality reduction was conducted (PCA and UMAP).

## 14,8K murine aging lung cells

We downloaded the raw count matrix and the metadata file from *Angelidis et al., 2019* from the Gene Expression Omnibus (accession number: GSE124872). We created an annData object with the raw count matrix and the available metadata (cell type annotation, age group, cluster, and mouse). We ran a standard processing pipeline: quality control, normalization and log-transformation of counts, selection of highly variable genes, batch-effect correction between individual mice using bbknn (*Park et al., 2018*), and dimensionality reduction (PCA and UMAP). In our analysis, we used the cell type annotations provided by the authors. We also annotated the rest of the murine aging lung datasets using their annotation as a reference. In order to do that, we computed the DEGs between each lung cell type and the rest of the dataset to obtain a set of gene markers for each cell type. We then used those markers to annotate the rest of the datasets using scoreCT (*Seninge, 2020*).

## 90,6K murine aging lung, spleen, and kidney cells

We downloaded the raw count matrices and the metadata files from *Kimmel et al., 2019* from the Gene Expression Omnibus (accession number: GSE132901). We selected lung samples (30,255 cells) and excluded kidney and spleen samples. We discarded the two samples from the individual Y1, as they showed a very different count distribution to the rest of the samples (see Appendix, Lack of evidence for increased transcriptional noise in aged tissues). We created an annData object with the count matrix and the metadata (sample, tissue, age, and mouse). We ran a standard processing pipeline: quality control, normalization and log-transformation of counts, highly variable gene selection, batch-effect correction between individual mice using bbknn (*Park et al., 2018*), dimensionality reduction (PCA and UMAP), and Leiden clustering (*Traag et al., 2019*) with high resolution value (`resolution=4`), so that we obtained a very granular clustering solution. We obtained 52 clusters, and we annotated them by projecting the cell type identity labels from the *Angelidis et al., 2019* dataset, using the automated cell type annotation tool scoreCT (*Seninge, 2020*). We checked that cells clustered primarily according to their cell type, meaning no important batch effects were present in the final datasets, and that clusters expressed the cell type markers expected according to their assigned cell type labels (see Appendix, Lack of evidence for increased transcriptional noise in aged tissues).

## 731 murine aging dermal fibroblasts

The count matrix and metadata from *Salzer et al., 2018* were downloaded from the Gene Expression Omnibus (accession number: GSE111136). A standard processing pipeline (quality control, normalization, log-transformation, HVG detection, PCA, neighbor computation, and UMAP dimensionality reduction) was applied to the dataset.

## 22,1K human aging skin cells

We downloaded raw count matrices from *Solé-Boldo et al., 2020* from the Gene Expression Omnibus (accession number: GSE130973). We ran a standard preprocessing pipeline on the count matrix: quality control, normalization, log-transformation, HVG detection, PCA, neighbor computation, and UMAP dimensionality reduction. We used the original cell type labels provided by the authors.

## Tabula Muris Senis lung datasets

The 3.2 K TMS FACS-sorted and the 4.4 TMS droplet lung cell datasets were downloaded from figshare. A standard preprocessing pipeline was run on the two datasets, and cluster labels were

harmonized with the rest of the murine aging lung datasets by using the genes differentially expressed between cell types from the Angelidis dataset as input for the automated cell-type annotation through scoreCT (*Seninge, 2020*).

## Human lung cell atlas

We downloaded the full lung and blood 10× dataset from the HLCA (*Travaglini et al., 2020*) from Synapse (ID: syn21041850). The original dataset consists of lung samples from three patients: a 46 years old male donor (donor 1), a 51 years old female donor (donor 2), and a 75 years old male donor (donor 3). The composition of the samples was not equivalent across donors: there were two samples from donor 1 (distal and medial), three samples from donor 2 (blood, distal, and proximal), and two samples from donor 3 (blood and distal). Thus, we selected the distal sample from the three donors and obtained a dataset of 18,542 cells from donor 1, 16,903 cells from donor 2, and 7524 cells from donor 3. We subsampled 7524 cells from each of the donors in order to correct for the age-group imbalance and obtained a dataset of 22,572 lung cells. We used this balanced dataset of distal samples from the three donors to create two datasets. On the one hand, we selected all lung cells from donors 1 and 3 (46 years old and 75 years old, all male) in order to create the 15,048 aging lung cell dataset used in the noise and enrichment analysis. On the other hand, we selected all alveolar macrophages from the three donors in order to create the 11,484 alveolar macrophage dataset.

## Human aging lung

We downloaded the mammalian aging lung dataset by *Raredon et al., 2019* from the Gene Expression Omnibus (accession number: GSE133747). The original dataset consists of human, pig, mouse, and rat samples. We selected human samples and ran the preprocessing and quality control pipeline on them: normalization, log-transformation, selection of highly variable genes, batch-effect correction between donors using harmony (*Korsunsky et al., 2019*), computation of the nearest neighbor graph, and Leiden clustering (*Traag et al., 2019*). The resulting dataset consisted of 17,867 cells from human male and female donors aged 21–88 years. We then projected the cell type labels from the human lung atlas onto the Raredon dataset by computing the DEGs between cell types in the human lung atlas dataset and using the first 300 DEGs to identify equivalent cell types in the Raredon dataset and projecting those onto the Raredon dataset using the unsupervised cell type annotation tool scoreCT (*Seninge, 2020*.) We identified 24 lung cell types from the HLCA. After using the cells from the 14 human donors in the annotation step, we selected a set of 9 donors in order to obtain a balanced aging dataset, using the following inclusion criteria: (1) donors contributing with very few cells were excluded (GSM4050113 and GSM4050107 consisted of 116 and 211 cells, respectively), (2) middle-aged donors were discarded in order to better explore the effects of aging, (3) donors were selected to ensure sex-stratification, and (4) we sought to obtain a balanced dataset in terms of age-group sizes. The final dataset consisted of 15,852 lung cells from 9 female and male human donors. We defined the age categories as young (21, 22, 32, 35, and 41 years old) and old (64, 65, 76, and 88 years old). The composition of the dataset was 7263 young (46%) and 8589 old cells (54%).

## Age-related change in transcriptional noise

To facilitate comparison with regard to cell type annotation, we harmonized the labels so that the four datasets were annotated using the cell identities originally defined by *Angelidis et al., 2019*. Then, we measured transcriptional noise as $1 - membership$ to cell type clusters in the young and old fractions of each dataset. We then measured the age-related difference in transcriptional noise per cell type by calculating the differences in median noise between the old and the young fraction for each lung cell type. In order to compare between the young and the old fraction of cells, each dataset was split into two datasets according to the age groups ('young' and 'old'), and the highly variable gene detection and dimensionality reduction (PCA, batch-corrected neighbor detection using harmony, and UMAP) steps where run again on each set of cells. Then, *Scallop* was run on each set of cells separately, using Leiden as the community detection method and using the following parameter values: `frac_cells = 0.8, n_trials = 30`. This was done on a range of resolution (`res`) values between 0.1 and 1.5, with a step of 0.1, and the membership scores obtained for each cell were averaged over all these resolution values in order to smooth the effect of clustering granularity on the membership scores.

We used the *freq* membership score, defined as the frequency of assignment of the most frequently assigned cluster label per cell.

### Age-related cell type enrichment

Changes in cell type abundance associated with aging were evaluated using binomial GLMs (*McCullagh and Nelder, 1989*). For each dataset, a binomial GLM was fitted to estimate the proportion of each cell type across all samples by treating each individual mouse as a replicate. First, the relative abundance of each cell type ($N\_ct$) and the relative abundance of the rest of the cell types taken together ($N\_other$) were computed. Then, a binomial GLM was fitted to these pairs of observations ($N\_ct$, $N\_other$) to estimate the proportions of cell types across samples by accounting for variation associated with sample origin (mouse) and to sample age (young vs old), and estimated marginal means (*Searle et al., 1980*) were computed using the R package *emmeans*. Odds ratios between 'Young' and 'Old' samples were computed for each cell identity.

### Code availability

The *Decibel* and *Scallop* repositories can be found at https://gitlab.com/olgaibanez/decibel and https://gitlab.com/olgaibanez/scallop, (copy archived at swh:1:rev:086edf77f471a-c0a786c2262b503842214d98357; *Ibañez-Solé, 2022b*), respectively. The official documentations sites can be found at https://scallop.readthedocs.io/en/latest/index.html and https://decibel.readthedocs.io/en/latest/index.html. Reproducible Jupyter notebooks with the analyses carried out in this study ar available at figshare (https://doi.org/10.6084/m9.figshare.20402817.v2).

## Acknowledgements

We thank Iñaki Inza for his thorough revision of the manuscript, Laura Yndriago for her feedback, and Sandra Fuertes for useful discussions. We thank Valentine Svensson for support with the application of GLMs to cell type abundance analysis of scRNAseq data. This work was supported by grants from Instituto de Salud Carlos III (AC17/00012, PI22/01247 and PI19/01621), cofunded by the European Union, and the 4D-HEALING project (ERA-Net program EracoSysMed, JTC-2 2017); Diputación Foral de Gipuzkoa; Ministry of Science and Innovation of Spain; and PID2020-119715GB-I00 funded by MCIN/AEI/10.13039/501100011033 and by "ERDF A way of making Europe". OI-S received the support of a fellowship from "la Caixa" Foundation (ID 100010434; code LCF/BQ/IN18/11660065), and from the European Union´s Horizon 2020 research and innovation programme under the Marie Skłodowska-Curie grant agreement No. 713673. AMA was supported by a Basque Government Postgraduate Diploma fellowship (PRE_2020_2_0081).

## Additional information

### Funding

| Funder | Grant reference number | Author |
|---|---|---|
| "la Caixa" Foundation | LCF/BQ/IN18/11660065 | Olga Ibañez-Solé |
| Instituto de Salud Carlos III | AC17/00012 | Ander Izeta<br>Marcos J Araúzo-Bravo |
| Instituto de Salud Carlos III | PI22/01247 and PI19/01621 | Ander Izeta |
| Ministerio de Ciencia e Innovación | PID2020-119715GB-I00 | Marcos J Araúzo-Bravo |
| European Regional Development Fund | MCIN/AEI/10.13039/501100011033 | Marcos J Araúzo-Bravo |
| H2020 Marie Skłodowska-Curie Actions | 713673 | Olga Ibañez-Solé |
| Eusko Jaurlaritza | PRE_2020_2_0081 | Alex M Ascensión |

| Funder | Grant reference number | Author |
|---|---|---|

The funders had no role in study design, data collection and interpretation, or the decision to submit the work for publication.

## Author contributions

Olga Ibañez-Solé, Conceptualization, Data curation, Software, Formal analysis, Funding acquisition, Investigation, Visualization, Methodology, Writing – original draft, Writing – review and editing; Alex M Ascensión, Conceptualization, Data curation, Software, Funding acquisition, Investigation, Visualization, Methodology, Writing – original draft, Writing – review and editing; Marcos J Araúzo-Bravo, Resources, Funding acquisition, Project administration, Writing – review and editing; Ander Izeta, Conceptualization, Supervision, Funding acquisition, Investigation, Writing – original draft, Project administration, Writing – review and editing

## Author ORCIDs

Olga Ibañez-Solé http://orcid.org/0000-0002-0552-9793
Alex M Ascensión http://orcid.org/0000-0002-0013-3052
Ander Izeta http://orcid.org/0000-0003-1879-7401

## Ethics

The authors declare no relevant ethical issues associated to this manuscript.

## Decision letter and Author response

Decision letter https://doi.org/10.7554/eLife.80380.sa1
Author response https://doi.org/10.7554/eLife.80380.sa2

# Additional files

## Supplementary files

• MDAR checklist

## Data availability

Code availability: The *Decibel* and *Scallop* repositories can be found at https://gitlab.com/olgaibanez/decibel (copy archived at swh:1:rev:8749a4e1ae05edcebb642fd7358a78b8468c511f) and https://gitlab.com/olgaibanez/scallop (copy archived at swh:1:rev:086edf77f471a-c0a786c2262b503842214d98357), respectively. The reproducible Jupyter notebooks with the analyses carried out in this study can be found in figshare (https://doi.org/10.6084/m9.figshare.20402817.v1).

The following dataset was generated:

| Author(s) | Year | Dataset title | Dataset URL | Database and Identifier |
|---|---|---|---|---|
| Ibañez-Solé O, Ascension AM | 2022 | Scallop_notebooks | https://doi.org/10.6084/m9.figshare.20402817.v1 | figshare, 10.6084/m9.figshare.20402817.v1 |

The following previously published datasets were used:

| Author(s) | Year | Dataset title | Dataset URL | Database and Identifier |
|---|---|---|---|---|
| Angelidis I, Simon LM, Schiller HB | 2019 | Multi-modal analysis of the aging mouse lung at cellular resolution | https://www.ncbi.nlm.nih.gov/geo/query/acc.cgi?acc=GSE124872 | NCBI Gene Expression Omnibus, GSE124872 |

*Continued*

| Author(s) | Year | Dataset title | Dataset URL | Database and Identifier |
|---|---|---|---|---|
| Kimmel JC, Penland L, Rubinstein ND, Hendrickson DG, Kelley DR, Rosenthal AZ | 2019 | A murine aging cell atlas reveals cell identity and tissue-specific trajectories of aging | https://www.ncbi.nlm.nih.gov/geo/query/acc.cgi?acc=GSE132901 | NCBI Gene Expression Omnibus, GSE132901 |
| Pisco A | 2020 | Tabula Muris Senis | https://figshare.com/articles/dataset/Tabula_Muris_Senis_Data_Objects/12654728 | Figshare, 12654728 |
| Travaglini KJ | 2020 | Human Lung Cell Atlas | https://doi.org/10.7303/syn21041850 | Synapse, 10.7303/syn21041850 |
| Raredon MS, Adams T, Schupp JC | 2019 | Single-cell connectomic analysis of adult mammalian lungs | https://www.ncbi.nlm.nih.gov/geo/query/acc.cgi?acc=GSE133747 | NCBI Gene Expression Omnibus, GSE133747 |
| Enge M, Arda E | 2017 | Single cell transcriptome analysis of human pancreas reveals transcriptional signatures of aging and somatic mutation patterns | https://www.ncbi.nlm.nih.gov/geo/query/acc.cgi?acc=GSE81547 | NCBI Gene Expression Omnibus, GSE81547 |
| Solé-Boldó L, Lyko F, Rodriguez-Pareses M | 2020 | Single-cell transcriptomes of the aging human skin reveal loss of fibroblast priming | https://www.ncbi.nlm.nih.gov/geo/query/acc.cgi?acc=GSE130973 | NCBI Gene Expression Omnibus, GSE130973 |
| Ximerakis M, Rubin LL, Lipnick SL, Levin J | 2019 | Single-cell transcriptomic profiling of the aging mouse brain | https://www.ncbi.nlm.nih.gov/geo/query/acc.cgi?acc=GSE129788 | NCBI Gene Expression Omnibus, GSE129788 |
| Heyn H, Rodriguez-Esteban G, Salzer MC | 2018 | scRNAseq dataset of murine aging dermal fibroblasts | https://www.ncbi.nlm.nih.gov/geo/query/acc.cgi?acc=GSE111136 | NCBI Gene Expression Omnibus, GSE111136 |
| Eling N | 2019 | scRNAseq dataset of murine aging T cells | https://zenodo.org/record/3522970#.Y2j8t77MJkg | Zenodo, 10.5281/zenodo.3522970 |

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

# Appendix 1

**Appendix 1—table 1.** Datasets used in the technical validation of *Scallop.*
Number of cells, number of genes, median number of genes per cell, and number of estimated cell populations in each dataset.

| Dataset | n_cells | n_genes | median_genes_per_cell | n_populations |
|---|---|---|---|---|
| PBMC3K | 2700 | 32738 | 817 | 12 |
| *Joost et al., 2016* | 1422 | 6410 | 1941 | 17 |
| *Paul et al., 2015* | 2730 | 3451 | 872 | 14 |
| *Moignard et al., 2015* | 3934 | 42 | 42 | 9 |
| Heart10K | 7713 | 11765 | 2035 | 26 |

# Appendix 2

**Appendix 2—table 1.** Seven single-cell RNA sequencing studies of different tissues where age-related increase in transcriptional noise was measured.

The number of cells (*N. cells*) in the table is the size of the dataset prior to quality control. The *Noise* column states whether an increase in transcriptional noise was reported in some/all cell types in the original articles. The *Scope* column summarizes the cell types where age-related increase in transcriptional noise was reported. The *Method* column specifies how transcriptional noise was measured in the original articles.

| Dataset | Tissue | Organism | N. cells | Noise | Scope | Method |
|---|---|---|---|---|---|---|
| *Enge et al., 2017* | Pancreas | Human | 2544 | Yes | In Beta cells. | (1) Biological over technicalvariation, (2) wholetranscriptome-based Euclidean distance to cell typemean, (3) invariant gene-based Euclidean distance to celltype mean. |
| *Martinez-Jimenez et al., 2019* | CD4+ T cells | Mouse | 1513 | Yes | Single cell type studied. | Percentage of cells expressingthe core activation program. |
| *Angelidis et al., 2019* | Lung | Mouse | 14,813 | Yes | In most cell types. | Distance to cell type mean. |
| *Kimmel et al., 2019* | Lung, spleen, kidney | Mouse | 30,255 30,512 29,815 | Yes | In many cell types. | (1) Overdispersion of genes, (2) invariant gene-based Euclidean distance to cell type mean, (3) whole transcriptome--based Manhattan distance to cell type mean. |
| *Ximerakis et al., 2019* | Brain | Mouse | 37,069 | No | Differences in magnitude and directionality between cell types. | Coefficient of variation of (1) all genes, (2) mitochondrial genes, (3) ribosomal genes. |
| *Salzer et al., 2018* | Dermal fibroblasts | Mouse | 731 | Yes | Single cell type studied. | Compactness of clusters on PCA plot. |
| *Solé-Boldo et al., 2020* | Skin | Human | 22,142 | Yes | In dermal fibroblasts. | Less clear GO (Gene Ontology) annotations. |

# Appendix 3

**Appendix 3—table 1.** Data inclusion criteria.

The general criteria for inclusion in the aging datasets used in this study was to include all samples from young and old individuals and to exclude newborn or pediatric individuals, as we did for the human pancreatic cell dataset (*Enge et al., 2017*) and the murine dermal fibroblast dataset (*Salzer et al., 2018*). Care was taken to make all aging datasets sex-balanced. This was not possible for some datasets, as they consisted of same-sex individuals. However, same-sex datasets were included in our study as sex could not be a confounding factor in the aging analysis.

| Dataset | Inclusion criteria | Number of individuals | Number of cells | Ages |
|---|---|---|---|---|
| *Enge et al., 2017* | All samples except those from pediatric individuals (0–6 years old) | Young: 2 Old: 3 | Young: 791 Old: 1023 | Young: 21 and 22 years Old: 38, 44, and 58 years |
| *Martinez-Jimenez et al., 2019* | Whole dataset. | Young: 9 Old: 12 | Young: 532 Old: 981 | Young: 3 months Old: 24 months |
| *Angelidis et al., 2019* | Whole dataset. | Young: 8 Old: 7 | Young: 7644 Old: 6526 | Young: 3 months Old: 24 months |
| *Kimmel et al., 2019* | Lung samples from all mice except Y1. | Young: 3 Old: 3 | Young: 13,352 Old: 12,998 | Young: 7 months Old: 24 months |
| *Ximerakis et al., 2019* | Whole dataset. | Young: 8 Old: 8 | Young: 16,028 Old: 21,041 | Young: 2–3 months Old: 21–23 months |
| *Salzer et al., 2018*. | All samples except those from newborn mice. | Young: 4 Old: 4 | Young: 329 Old: 332 | Young: 2 months Old: 18 months |
| *Solé-Boldo et al., 2020* | Whole dataset. | Young: 2 Old: 3 | Young: 8316 Old: 13,826 | Young: 25 and 27 years Old: 53, 69, and 70 years |

# Appendix 4

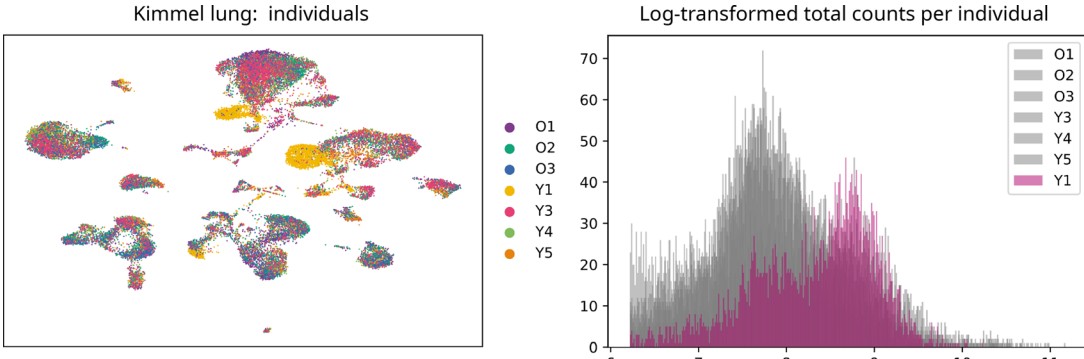

**Appendix 4—figure 1.** Uniform manifold approximation and projection plot showing the samples from the seven individuals present in the Kimmel lung dataset. Even though most cells cluster together according to their cell type rather than by individual, samples from donor Y1 cluster together. We observed that there was a big batch effect between this and the rest of the individuals. Histogram showing the log-transformed total number of counts/cell per individual mice. The distribution of counts/cell of the samples from mouse Y1 is very different to the rest of the samples. This difference could not be overcome using the batch-effect correction tool *bbknn*. Downsampling the counts so that the number of counts/cell was balanced across individual mice did not solve the problem either. Therefore, we decided to discard the samples Y1L1 and Y1L2.

# Appendix 5

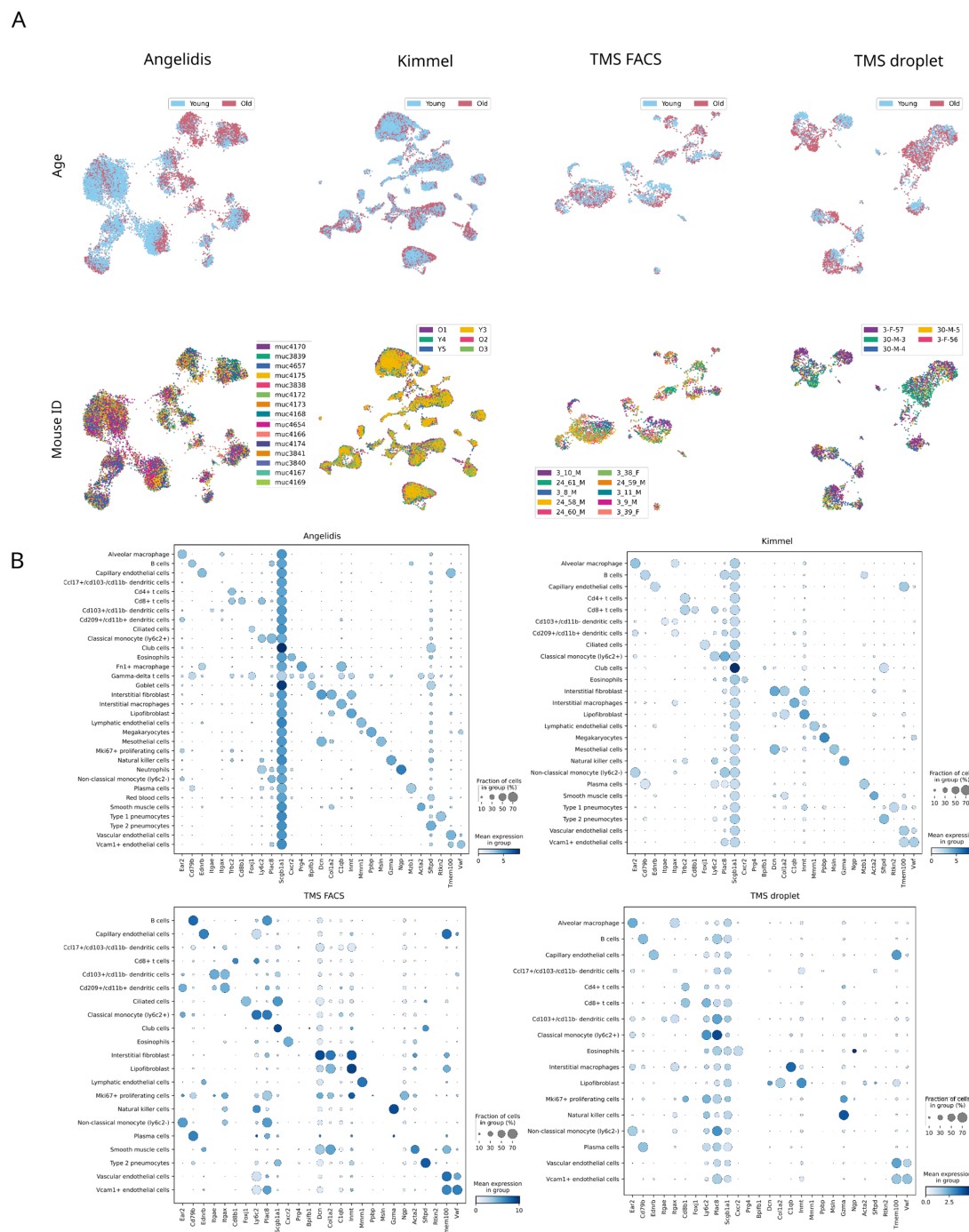

**Appendix 5—figure 1.** Murine aging lung datasets: sample composition and cell type marker expression. (**A**) There are no mouse- or age-related batch effects. Uniform manifold approximation and projection plots of the four aging lung datasets showing the age and mouse labels. Cells cluster according to their cell type rather than to their age group or individual mouse. (**B**) Expression of lung cell type markers by each annotated cluster. The dotplots show the expression of the cell type markers from ***Angelidis et al., 2019*** on the four annotated lung datasets. The size of the dots represents the fraction of cells expressing one particular marker in the group of cells assigned a particular cell type label. The color represents the level of expression of the marker in that group averaged over the cells that have a positive expression of that marker.

# Appendix 6

**Appendix 6—table 1.** Number of cells, sex, and age composition of the human aging lung datasets.

| Dataset | No. of young cells | No. of old cells | No. of young donors | No. of old donors | Total cells |
|---|---|---|---|---|---|
| *Raredon et al., 2019* | 7263 | 8589 | Three females: 21, 32, 41, years old Two males: 22, 35 years old | Two females: 76, 88 years old Two males: 64, 65 years old | 15,852 |
| *Travaglini et al., 2020* (HLCA) | 7524 | 7524 | One male: 46 years old | One male: 75 years old | 15,048 |

