## [Editor Report]

The authors present an important perspective surrounding a fundamental question of associations between transcriptional noise and the aging process. They develop new methods to probe stochastic gene expression from single-cell sequencing data where their results suggest that associations between noise and age can be attributed to alternative metrics such as shifts in cellular identity. These methods and analyses provide an important framework to guide the fields of gene expression regulation and aging.

---

## [Decision Letter]

**Decision letter after peer review:**

Thank you for submitting your article "Lack of evidence for increased transcriptional noise in aged tissues" for consideration by *eLife*. Your article has been reviewed by 3 peer reviewers, and the evaluation has been overseen by a Reviewing Editor and David James as the Senior Editor. The reviewers have opted to remain anonymous.

Essential revisions:

All 3 reviewers find the analyses and conclusions timely and of broad interest. Specifically, the observation that lack of noise is associated with aging is intriguing. While your approach (*Scallop*) has the potential to be of wide use and interest, the robustness of the pipeline and systematic evaluation should be expanded in more detail. We judge that this would make your Python package more likely to be used for analysis of transcriptional noise in other systems, and, importantly, substantially strengthen the conclusions made in this study. You can see from the comments in the Public Review that all 3 reviewers are supportive overall but list specific suggestions for potential ways to improve. Obviously if you have additional questions about these, feel free to reach out.

*Reviewer #2 (Recommendations for the authors):*

1. The authors seem to have downloaded count matrices for published datasets. Were they all preprocessed in the same way? Can authors rule out different pre-processing steps affecting the inconsistent results between studies?

2. Overall, I found the comparison across methods particularly interesting, but the authors seem to overlook the possibility that they might capture different biology and search for consistency. I would be interested in a discussion point on what each method can measure biologically.

*Reviewer #3 (Recommendations for the authors):*

Despite the weaknesses mentioned in the Public Review, this is still an important study and I would support the publication of this manuscript in *eLife* if it is sufficiently revised.

[Editors' note: further revisions were suggested prior to acceptance, as described below.]

Thank you for resubmitting your work entitled "Lack of evidence for increased transcriptional noise in aged tissues" for further consideration by *eLife*. Your revised article has been evaluated by David James (Senior Editor) and a Reviewing Editor. We are prepared to consider a revised submission detailing a few minor clarifications.

In particular, all of us found the manuscript to be significantly improved and, with the addition of some minor changes, ready for publication. These include updating the figshare repository code to reflect current manuscript results and several text-based revisions regarding interpretations of transcriptional noise during identity loss and appropriate referencing of previous studies. These are detailed below and we look forward to moving forward with acceptance. Below you will find related comments made by both reviewers:*Reviewer #1:*

1) The figshare repository with the code does not seem to be updated. It's important that the code related to the generation of synthetic data and their analysis is uploaded.

*Reviewer #2:*

The revised manuscript is stronger than the original submission. The authors either directly address the concerns or specifically acknowledge what was/is a weak point of the study (for valid reasons related to the technique itself).

There are indeed very few longitudinal studies that focused on cellular aging-noise connection in live cells, therefore additional work is needed to quantitatively/experimentally sort out the contributions of intrinsic/extrinsic factors to the overall transciptional variability. In the Discussion section where the authors discuss the few longitunidal examples (Liu et al. and Sarnoski et al. papers), the current writing gives the impression that no mechanism has been proposed or studied from experimental and/or computational perspective to explain the mechanism of noise-change during aging. Actually, based on stochastic simulations matching experimental results, both of the above papers have already proposed that the observed intracellular variability dynamics in aging haploid/diploid yeast could be due to specific stochastic promoter state transition rates occurring during single-cell aging. While the age-associated changes in chromatin remodeling is just one mechanism (among potentially several) that could explain intrinsic noise dynamics during aging, it is still important and should be appropriately acknowledged in the same discussion paragraph.

---

## [Author Response]

Essential revisions:All 3 reviewers find the analyses and conclusions timely and of broad interest. Specifically, the observation that lack of noise is associated with aging is intriguing. While your approach (*Scallop*) has the potential to be of wide use and interest, the robustness of the pipeline and systematic evaluation should be expanded in more detail. We judge that this would make your Python package more likely to be used for analysis of transcriptional noise in other systems, and, importantly, substantially strengthen the conclusions made in this study. You can see from the comments in the Public Review that all 3 reviewers are supportive overall but list specific suggestions for potential ways to improve. Obviously if you have additional questions about these, feel free to reach out.

We thank the editors and reviewers for a fair and thorough review of the manuscript and their constructive suggestions to improve it. We have now included novel data that substantially increase the evidence on our algorithm’s robustness as well as provided new examples of where previous attempts to measure age-associated transcriptional noise provided inconclusive/inaccurate results. The new results strengthen our original conclusions and may facilitate wider use of the pipeline for other biological systems as suggested.

Reviewer #2 (Recommendations for the authors):1. The authors seem to have downloaded count matrices for published datasets. Were they all preprocessed in the same way? Can authors rule out different pre-processing steps affecting the inconsistent results between studies?

We agree with the reviewer that the preprocessing and quality control of RNAseq datasets can have a great impact on the output of downstream analysis. This is why we made all of our comparisons between age groups of the same dataset, to ensure that the preprocessing pipeline was not a confounding factor in our analysis. This way, even though individual datasets might have been preprocessed differently, the comparisons are always made between samples that have been processed together in identical experimental conditions and using the same computational pipeline, the age-difference of the donors/mice being the sole difference between them.

2. Overall, I found the comparison across methods particularly interesting, but the authors seem to overlook the possibility that they might capture different biology and search for consistency. I would be interested in a discussion point on what each method can measure biologically.

We think this is a very interesting point, and believe that the output of the different existing methods is certainly influenced by different sources of both technical and biological variability. For instance, the ERCC-based method proposed by Enge et al. is likely the only method that explicitly accounts for and removes the effect of technical noise, and methods that measure the dispersion of gene expression (standard deviation of the expression, coefficient of variation, etc) are likely to be more sensitive to changes in the expression of highly expressed genes, and so on. However, we think that the main contribution of our work is to show that, regardless of the method used to measure transcriptional noise, there is no convincing evidence that increased noise (of any type) is a universal feature of aged cells. We fully agree that other methods may be capturing different biology but finding out the underlying cause of apparent “noise” in each dataset is certainly out of scope for this manuscript.

[Editors' note: further revisions were suggested prior to acceptance, as described below.]

Reviewer #1:1) The figshare repository with the code does not seem to be updated. It's important that the code related to the generation of synthetic data and their analysis is uploaded.

Thank you very much for noting this omission. The Jupyter notebooks used to create the Figure 1 – Supplements 1-6 and the Figure 5- Supplement 2 have been added to the latest version of the figshare repository:

(https://figshare.com/articles/software/Scallop_notebooks/20402817).

Reviewer #2:The revised manuscript is stronger than the original submission. The authors either directly address the concerns or specifically acknowledge what was/is a weak point of the study (for valid reasons related to the technique itself).

Thank you very much.

There are indeed very few longitudinal studies that focused on cellular aging-noise connection in live cells, therefore additional work is needed to quantitatively/experimentally sort out the contributions of intrinsic/extrinsic factors to the overall transciptional variability. In the Discussion section where the authors discuss the few longitunidal examples (Liu et al. and Sarnoski et al. papers), the current writing gives the impression that no mechanism has been proposed or studied from experimental and/or computational perspective to explain the mechanism of noise-change during aging. Actually, based on stochastic simulations matching experimental results, both of the above papers have already proposed that the observed intracellular variability dynamics in aging haploid/diploid yeast could be due to specific stochastic promoter state transition rates occurring during single-cell aging. While the age-associated changes in chromatin remodeling is just one mechanism (among potentially several) that could explain intrinsic noise dynamics during aging, it is still important and should be appropriately acknowledged in the same discussion paragraph.

We thank the reviewer for pointing out that we failed to convey that a mechanism had been proposed for the emergence of transcriptional noise with aging. This is now corrected as follows:

“In any case, the origin of transcriptional noise is unclear, as it could arise from many different sources. Most importantly, it not possible to distinguish between intrinsic and extrinsic noise from a snapshot of cellular states, i.e., one cannot tell whether the observed differences between cells in a single-cell RNA experiment reflect time-dependent variations in gene expression or differences between cells across a population (Ham et al. 2021). Interestingly, recent work by Liu et al. measuring intrinsic noise in *S. cerevisiae* showed that aging is associated with a steady decrease in noise, with a sudden increase in soon-to-die cells. They proposed that noise decreasing in normal aging was due to increased rates in chromatin state transitions, and showed that disrupting chromatin structure at the gene promoter was sufficient to change this course. Another longitudinal study found an increase in extrinsic noise and a lack of change in intrinsic noise in diploid yeast Sarnoski et al. (2018).”